# Plant SYP12 syntaxins mediate an evolutionarily conserved general immunity to filamentous pathogens

Hector M Rubiato[1†‡], Mengqi Liu[1], Richard J O'Connell[2], Mads E Nielsen[1*†]

[1]University of Copenhagen, Faculty of Science, CPSC, Department of Plant and Environmental Sciences, Copenhagen, Denmark; [2]University of Paris-Saclay, INRAE, UR BIOGER, Thiverval-Grignon, France

**Abstract** Filamentous fungal and oomycete plant pathogens that invade by direct penetration through the leaf epidermal cell wall cause devastating plant diseases. Plant preinvasive immunity toward nonadapted filamentous pathogens is highly effective and durable. Pre- and postinvasive immunity correlates with the formation of evolutionarily conserved and cell-autonomous cell wall structures, named papillae and encasements, respectively. Yet, it is still unresolved how papillae/encasements are formed and whether these defense structures prevent pathogen ingress. Here, we show that in *Arabidopsis* the two closely related members of the SYP12 clade of syntaxins (PEN1 and SYP122) are indispensable for the formation of papillae and encasements. Moreover, loss-of-function mutants were hampered in preinvasive immunity toward a range of phylogenetically distant nonadapted filamentous pathogens, underlining the versatility and efficacy of this defense. Complementation studies using SYP12s from the early diverging land plant, *Marchantia polymorpha*, showed that the SYP12 clade immunity function has survived 470 million years of independent evolution. These results suggest that ancestral land plants evolved the SYP12 clade to provide a broad and durable preinvasive immunity to facilitate their life on land and pave the way to a better understanding of how adapted pathogens overcome this ubiquitous plant defense strategy.

**\*For correspondence:**
maen@plen.ku.dk

†These authors contributed equally to this work

**Present address:** ‡Centre for Plant Biotechnology and Genomics (UPM-INIA), Universidad 11 Politécnica de Madrid Campus de Montegancedo, Pozuelo de Alarcón, Madrid, Spain

**Competing interest:** The authors declare that no competing interests exist.

## Editor's evaluation

The study provides evidence that PEN1 and SYP122 regulate structures important for defense against filamentous pathogen infection. The formation of papillae and encasement of haustoria both appear to be ancient defense mechanisms in land plants. The findings that PEN1 and its close homolog SYP122 play an overlapping role in pre- and postinvasive immunity against cell-wall-penetrating filamentous pathogens advance our understanding of defense mechanisms against filamentous pathogens.

## Introduction

In response to attack by filamentous pathogens, plants assemble localized preinvasive papillae and postinvasive encasements at sites of attempted cell entry (*Figure 1—figure supplement 1A and B*; *Hückelhoven and Panstruga, 2011*). These conserved defense structures are thought to contain antimicrobial cargo that provide effective and durable immunity against diverse filamentous pathogens, and fossil evidence suggests that these structures appeared very early in the evolution of land plants (*O'Connell and Panstruga, 2006*; *Krings et al., 2007*; *Overdijk et al., 2016*; *Hansen and Nielsen, 2017*). However, although their frequent association with resistance to nonadapted pathogens is well-known, direct evidence that papillae/encasements contribute to immunity against these

pathogens is still lacking. The discovery that the secretory syntaxin PEN1 is required for the timely formation of papillae to hamper penetration by nonadapted powdery mildew fungi highlighted the key role played by membrane trafficking in plant immunity (*Collins et al., 2003*; *Nielsen et al., 2012*; *Nielsen and Thordal-Christensen, 2013*). PEN1 (also referred to as syntaxin of plants 121 or SYP121) is a Qa-SNARE primarily located at the plasma membrane. On the other hand, the ARF-GEF, GNOM, regulates recycling of PEN1 between the plasma membrane and the *trans*-Golgi network, where it is needed for a fast papilla response (*Nielsen et al., 2012*). Accordingly, PEN1 is thought to mediate membrane fusion both at the plasma membrane and at the *trans*-Golgi network. The ROR2 syntaxin in barley is orthologous to PEN1 (*Collins et al., 2003*), suggesting that the preinvasive immunity dependent on PEN1 or ROR2 was present in early angiosperms before the divergence of monocots and dicots, spanning some 140 million years (My) of evolution. Focal accumulation of GFP-PEN1 at attack sites in response to *Blumeria graminis* f.sp. *hordei* (*Bgh*) is also seen for ROR2 in barley and highlights a likely conserved functionality in plant defense against powdery mildews (*Bhat et al., 2005*). The strong extracellular accumulation of GFP-PEN1 signal localized to fungal attack sites can be explained by the secretion of extracellular vesicles, also known as exosomes, which accumulate to a high degree in both papillae and encasements (*An et al., 2006*; *Meyer et al., 2009*; *Nielsen et al., 2012*). While exosomes are suggested to contain miRNAs directed against the invading pathogen, their topology makes it difficult to associate syntaxin functionality (*Meyer et al., 2009*; *Hansen and Nielsen, 2017*). GFP-PEN1 is generally considered a valid exosomal marker, yet the biogenesis pathway of PEN1-labeled exosomes remains unclear (*Rutter and Innes, 2017*). Loss of PEN1 or ROR2 delays, but does not prevent, the papilla response upon attack by the barley powdery mildew fungus *Bgh* in *Arabidopsis* or barley, respectively (*Assaad et al., 2004*; *Böhlenius et al., 2010*). In contrast, the deposition of encasements around developing intracellular pathogenic structures (IPS) is unaffected by the loss of PEN1 (*Wen et al., 2011*; *Nielsen et al., 2017*). Furthermore, although many other filamentous pathogens also penetrate directly through the plant epidermal cell wall, the PEN1- and ROR2-dependent immunity appears dedicated toward the haustorium-forming powdery mildew and rust fungi (*Collins et al., 2003*; *Jarosch et al., 2005*; *Lipka et al., 2005*; *Loehrer et al., 2008*; *Hoefle et al., 2009*). Thus, the molecular mechanisms underlying papilla/encasement formation, as well as their role in general plant pre- and postinvasive immunity toward other filamentous pathogens, remain obscure.

PEN1 and ROR2 belong to the SYP12 clade of secretory syntaxins, which first appears in embryophytes and is conserved in all land plants (*Slane et al., 2017*). Consequently, the appearance of the SYP12 clade was proposed to play a key role in the terrestrialization of plants by enabling specialization of the secretory pathway (*Sanderfoot, 2007*). This is supported by the fact that four of the five SYP12 clade members in *Arabidopsis* function in specialized forms of polarized secretion, these being preinvasive immunity toward powdery mildew fungi (PEN1), root hair tip growth (SYP123), and pollen tube tip growth (SYP124 and SYP125) (*Collins et al., 2003*; *Slane et al., 2017*; *Ichikawa et al., 2014*). Recently, PEN1 and SYP122, which is the closest homologue of PEN1, were suggested to mediate distinct but complementary secretory pathways during vegetative plant growth (*Waghmare et al., 2018*). Indeed, *syp122* mutants show an altered cell wall composition, but in contrast to *pen1* mutants, there is no effect on the preinvasive immunity toward *Bgh* (*Assaad et al., 2004*). Interestingly, the *pen1 syp122* double mutant is autoimmune, developing a severe necrotic phenotype and accumulating the defense hormone salicylic acid (*Assaad et al., 2004*; *Zhang et al., 2007*; *Zhang et al., 2008*). However, prior to the onset of necrosis, *pen1 syp122* double mutant plants are indistinguishable from wild-type plants. This underlines that PEN1 and SYP122 likely mediate a highly specialized form of secretion, whereas general secretion relies on the evolutionarily ancient Qa-SNARE SYP132 (*Park et al., 2018*; *Karnahl et al., 2018*). The development of necrotic lesions in *pen1 syp122* double mutant plants is markedly delayed when the accumulation of salicylic acid is hampered (*Zhang et al., 2007*; *Zhang et al., 2008*). Accordingly, the *pen1 syp122* double mutant autoimmune phenotype was suggested to result from activation of nucleotide-binding leucine-rich repeat receptors (NLRs), immune sensors that recognize pathogen effectors, directly or indirectly, which leads to a programmed cell death response (*Wang et al., 2019a*; *Wang et al., 2019b*; *Horsefield et al., 2019*; *Wan et al., 2019*). In support of this, mutations in the deubiquitinase AMSH3 that impairs the cell death reaction caused by activation of coiled-coil (CC)-NLRs also suppress the autoimmune phenotype of the *pen1 syp122* double mutant (*Schultz-Larsen et al., 2018*). The autoimmune phenotype shows that while PEN1 and SYP122 are likely to have separate functions in the preinvasive

immunity toward *Bgh*, the two syntaxins also share an as-yet unresolved function that is likely related to defense and not general development.

Here, we show that the formation of papillae and encasements in *Arabidopsis* requires either of the two SYP12 clade members, PEN1 and SYP122, because plants lacking both syntaxins are completely devoid of both types of defense structure. We show that PEN1 and SYP122 provide pre- and postinvasive immunity not only against the powdery mildew *Bgh* but also against distantly related filamentous pathogens that invade host epidermal cells, namely, the fungus *Colletotrichum destructivum* and the oomycete *Phytophthora infestans*. Complementation studies show that SYP12A from *Marchantia polymorpha* functions and behaves like SYP122 in *Arabidopsis*, thereby revealing a conserved ancestral mechanism. We conclude that papilla and encasement formation is mediated by a specialized function of the SYP12 clade of syntaxins and that these defense structures correlate with a broad and effective resistance toward filamentous pathogens that penetrate the epidermal cell layer. Moreover, we suggest that early land plants evolved the SYP12 clade of secretory syntaxins to provide a durable immunity that helped plants move on to land and possibly shape mutualistic plant-fungus interactions.

## Results

The role of PEN1 in timely papilla deposition in response to *Bgh* attack prompted us to investigate the involvement of other syntaxins in preinvasive immunity, notably SYP122 that shares a currently unknown function with PEN1. This shared function is apparent from the autoimmune lesion-mimic phenotype of the *pen1 syp122* double mutant (*Assaad et al., 2004*; *Zhang et al., 2007*; *Zhang et al., 2008*). Yet, during the first ~2 weeks of growth, the *pen1 syp122* mutant develops no visible symptoms, leaving a short window in which to study the interaction with *Bgh* before the onset of necrosis. From analysis of unattacked plants, we found that the *pen1 syp122* mutant developed spontaneous callose deposits despite having a wild-type-like appearance (*Figure 1A and B*). Callose is a polysaccharide (β-1,3-glucan) secreted in response to various biotic and abiotic stresses and is a major constituent of papillae and encasements. Although the deposition of callose is not required to block the penetration event by *Bgh* per se, it serves as a reliable visual marker for easy detection of papillae and encasements (*Ellinger et al., 2013*). We found that in response to attack by *Bgh* plants lacking PEN1 and SYP122 were unable to form papillae at sites of unsuccessful attack by fungal appressoria, as visualized by detection of callose (*Figure 1C and D*). Surprisingly, while the appressorial germ tube did not induce a papilla response in *pen1 syp122* mutants, we observed large callosic deposits induced by the noninvasive primary germ tube, which were not observed in any other genetic background (*Figure 1D, F, and G*, *Figure 1—figure supplement 1A*). Thus, SYP122 seems to play a complementary role to PEN1 in papilla formation upon attack by *Bgh*. However, despite being unable to form a papilla at the attempted entry site, the *pen1 syp122* mutant remains capable of generating focused depositions of callose elsewhere and seems to be highly sensitive to signals released by *Bgh*.

In response to a successful penetration attempt by *Bgh*, the host cell initiates postinvasive immunity, where an encasement forms as an extension of the penetrated papilla (*Figure 1—figure supplement 1B*). The encasement expands and, eventually, completely encloses the IPS and likely prevents the pathogen from proliferating. Previous observations have shown that although *pen1* mutants display a delayed papilla formation response to attack by *Bgh*, these plants seem unaffected in their ability to form encasements when successfully penetrated (*Assaad et al., 2004*; *Wen et al., 2011*; *Nielsen et al., 2017*). In contrast to our observations of unsuccessful penetration sites, the successful penetration of *pen1 syp122* cells often induced a strong but diffuse, unfocused accumulation of callose at the fungal entry site (*Figure 1E–G*, *Figure 1—video 1*, *Figure 1—video 2*). Interestingly, we did not observe any encasement-like structures. Instead, the callosic deposits in the *pen1 syp122* mutant were superficial and did not extend inward from the outer surface of the host cell enveloping the growing IPS. This would suggest that the penetration event by *Bgh* triggers a strong callose response, but formation of the encasement is dependent on a functional overlap of the PEN1 and SYP122 syntaxins. Furthermore, we tested the adapted powdery mildew *Golovinomyces orontii*, which also induces encasement formation, although at a later stage and less frequently than *Bgh*. For *G. orontii*, IPS formation seemed unaffected in *pen1 syp122* and no encasements were found. Interestingly, successful penetration by *G. orontii* did not induce the strong, diffuse callose response described for *Bgh* (*Figure 1—figure supplement 1C and D*, *Figure 1—video 3*, *Figure 1—video 4*).

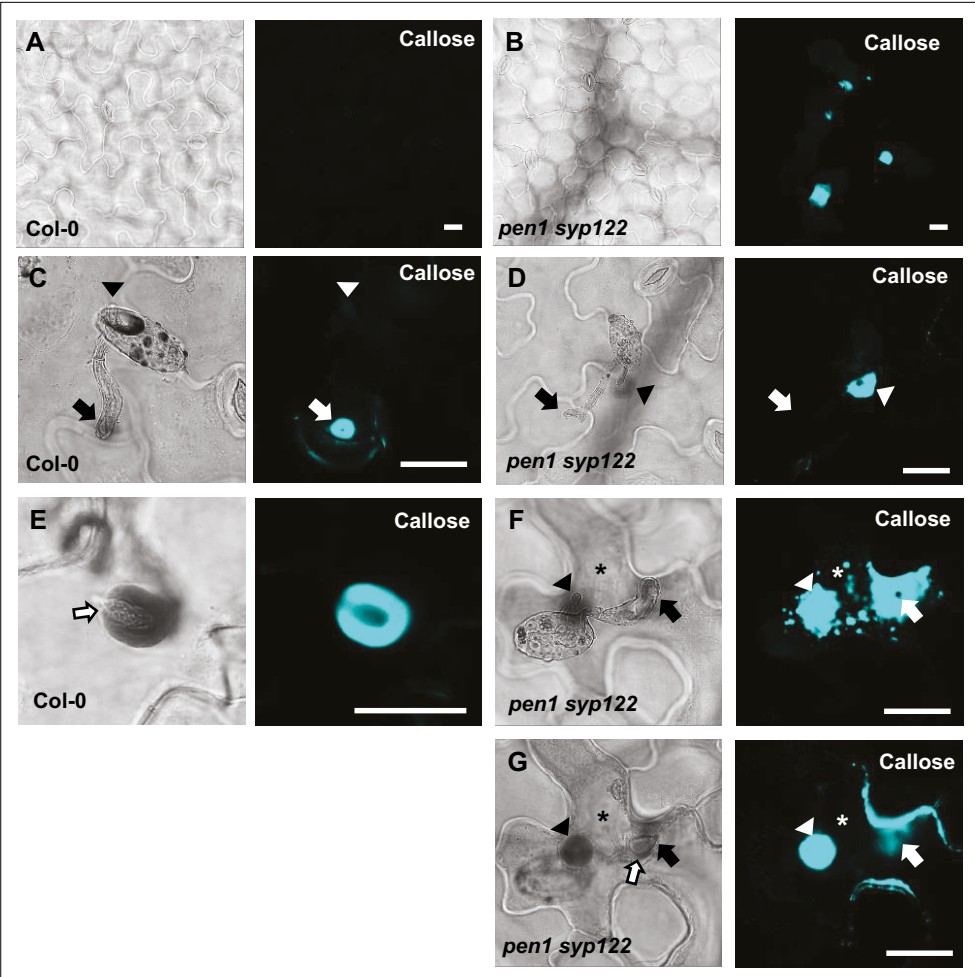

**Figure 1.** PEN1/SYP122 are required for papilla and encasement responses to *Bgh*. (**A, B**) Spontaneous callose depositions in leaves of 3-week-old plants. (**C–G**) Accumulation of callose in response to *Bgh* attack at appressoria (arrows) and primary germ tube (arrowheads) in (**C, D**), nonpenetrated cells and (**E–G**) cells following a successful penetration. Open arrows point to the developing intracellular pathogenic structure (IPS). (**G**) Same as (**F**), but optical section taken below the point of entry. * marks cells with cell death response. Bars = 20 μm.

The online version of this article includes the following video and figure supplement(s) for figure 1:

**Figure supplement 1.** Pre- and postinvasive immunity.

**Figure 1—video 1.** Z-stack (callose and bright-field overlay) of successful penetration by *Bgh* in Col-0.
https://elifesciences.org/articles/73487/figures#fig1video1

**Figure 1—video 2.** Z-stack (callose and bright-field overlay) of successful penetration by *Bgh* in *pen1 syp122*.
https://elifesciences.org/articles/73487/figures#fig1video2

**Figure 1—video 3.** Z-stack (callose and bright-field overlay) of successful penetration by *G. orontii* in Col-0.
https://elifesciences.org/articles/73487/figures#fig1video3

**Figure 1—video 4.** Z-stack (callose and bright-field overlay) of successful penetration by *G. orontii* in *pen1 syp122*.
https://elifesciences.org/articles/73487/figures#fig1video4

The lack of papillae and encasements in the *pen1 syp122* mutant prompted us to investigate the secretion of exosomes in response to *Bgh* attack. We speculated that the plant exosomal marker TET8-GFP (TETRASPANIN 8), which accumulates at sites of attack by *Botrytis*, would also accumulate in response to attack by *Bgh* (*Boavida et al., 2013*; *Cai et al., 2018*). In unattacked epidermal cells of wild-type plants, we found TET8-GFP localized mainly on the plasma membrane (*Figure 2A*). TET8-GFP signal focally accumulated in response to successful penetration by *Bgh* and seems a reliable marker for the encasement (*Figure 2B*). In unattacked epidermal cells of the *pen1 syp122*

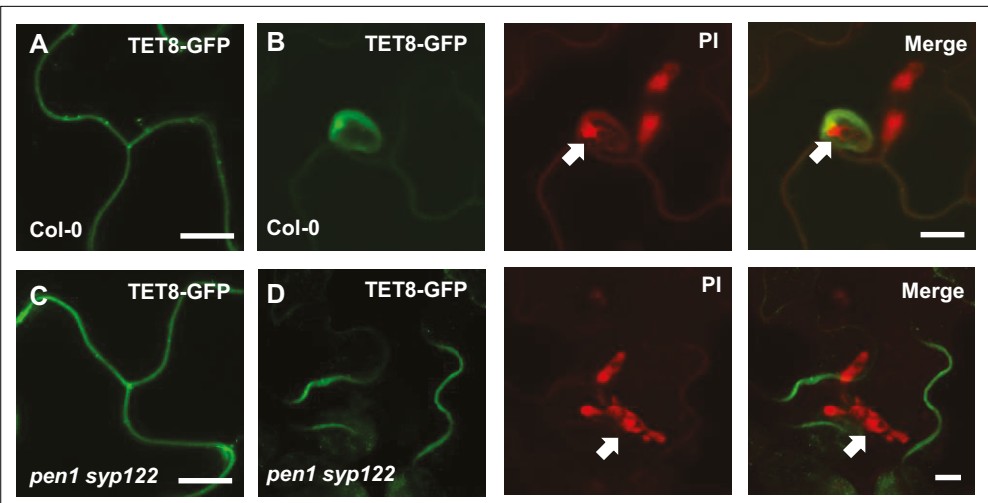

**Figure 2.** PEN1/SYP122 are required for accumulation of TET8-GFP in response to penetration by *Bgh*. (**A–D**) Localization of TET8-GFP in (**A, C**) resting epidermal cells and in (**B, D**) response to successful penetration by *Bgh* (stained with propidium iodide [PI]) and initiation of the intracellular pathogenic structure (IPS) (arrows). Bars = 10 µm.

The online version of this article includes the following video for figure 2:

**Figure 2—video 1.** Z-stack (GFP and propidium iodide overlay) of successful penetration by *Bgh* in Col-0.
https://elifesciences.org/articles/73487/figures#fig2video1

**Figure 2—video 2.** Z-stack (GFP and propidium iodide overlay) of successful penetration by *Bgh* in *pen1 syp122*.
https://elifesciences.org/articles/73487/figures#fig2video2

mutant, the TET8-GFP localization was similar to that seen in wild-type plants, whereas successful penetration by *Bgh* did not induce focal accumulation of TET8-GFP (*Figure 2C and D*, *Figure 2—video 1*, *Figure 2—video 2*). Thus, the general localization of TET8-GFP was not affected in the *pen1 syp122* mutant. Only in the event of fungal attack where TET8-GFP is transported to the infection site in wild-type plants this did not occur in the double mutant, supporting the idea that PEN1 and SYP122 play a specific role in pre- and postinvasive immunity.

We noticed that the spontaneous callose deposits formed in the lesion-mimic *pen1 syp122* mutant made it difficult to compare pre- and postinvasive immunity responses during attack by *Bgh*. Furthermore, successful penetration of epidermal cells in *pen1 syp122* by *Bgh*, and even *G. orontii*, resulted in a fast-developing cell death response that would likely compromise encasement formation. To overcome the difficulties of working with the *pen1 syp122* mutant, we made use of the partially rescued triple mutant, *fmo1 pen1 syp122* (*Zhang et al., 2008*). Flavin-dependent-monooxygenase1 (FMO1) enables *Arabidopsis* plants to generate N-hydroxypipecolic acid, which plays a central role in pathogen-inducible plant immunity (*Hartmann et al., 2018*). Similar to introducing a number of SA-signaling mutations, the loss of FMO1 clearly relieves the downstream effects of the *pen1 syp122* lesion-mimic mutant (*Figure 3—figure supplement 1A and B*). However, importantly, these mutations do not affect preinvasive immunity toward *Bgh*, thereby clearly separating the two phenotypes mechanistically (*Zhang et al., 2007* and this work). Moreover, *fmo1 pen1 syp122* plants did not display the large callosic deposits induced by the noninvasive primary germ tube. To ensure that the autoimmune phenotype of *fmo1 pen1 syp122* would not compromise fungal development, we first tested *G. orontii*. During the initial 48 hr after inoculation, we found that formation of the IPS and secondary hyphae in *fmo1 pen1 syp122* was indistinguishable from the *fmo1* control plants (*Figure 3—figure supplement 1C and D*). In response to attack by *Bgh*, we found that *fmo1* plants formed papillae at the site of attack that were identical to wild-type as visualized by detection of callose (*Figure 3A–C*, *Figure 3—figure supplement 2A–D*). Instead, plants lacking PEN1 and SYP122 were unable to form discrete papillae at sites of unsuccessful fungal attack. To support these findings, we also tested the ability of these plants to accumulate papillary $H_2O_2$ in response to attack by *Bgh* using DAB (*Thordal-Christensen et al., 1997*). Similar to the staining of callose, the accumulation of $H_2O_2$ is used as a

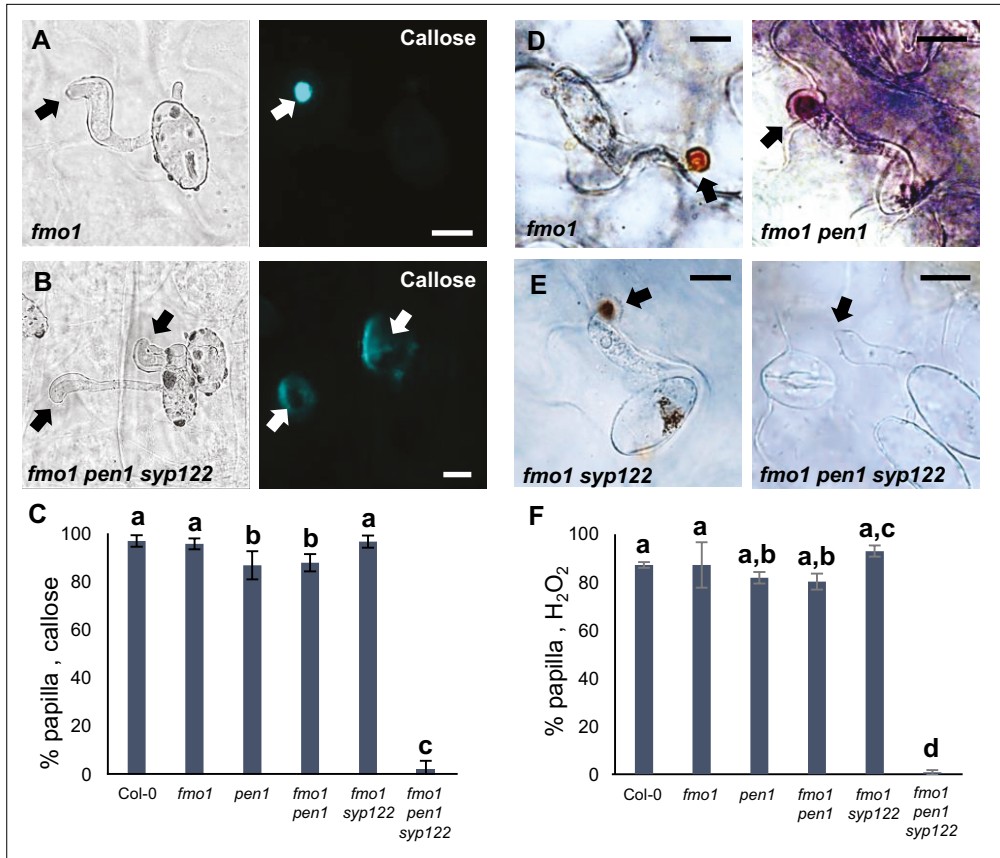

**Figure 3.** PEN1/SYP122 are required for papilla responses toward *Bgh*. (**A, B, D, E**) Accumulation of callose (**A, B**) or $H_2O_2$ (**D, E**) in response to *Bgh* attack (arrows) at nonpenetrated attack sites. Bars = 10 μm. (**C, F**) Frequency of papillae in response to *Bgh* in nonpenetrated cells as detected by staining for callose (**C**) or $H_2O_2$ (**F**). (**C, F**) All values are mean ± SD (n = 5 leaves per genotype). Different letters indicate significantly different values at p≤0.001 estimated using logistic regression.

The online version of this article includes the following source data and figure supplement(s) for figure 3:

**Source data 1.** Source data for the graphs in *Figure 3C and F*.

**Figure supplement 1.** Loss of FMO1 attenuates *pen1 syp122* autoimmunity.

**Figure supplement 2.** PEN1/SYP122 are required for papilla responses to *Bgh*.

convenient marker of the defense structure. In accordance with our observations on callose deposition, we found that *fmo1 pen1 syp122* plants failed to accumulate papillary $H_2O_2$ (*Figure 3D–F*). Thus, these results confirmed that plants lacking PEN1 and SYP122 failed to form papillae at sites of attack by *Bgh*.

Similar to the papilla response, we found that *fmo1* mutants reacted like wild-type plants in response to successful penetration by *Bgh*, as visualized by the detection of callose encasements around the IPS (*Figure 4A–C*, *Figure 4—figure supplement 1A–D*). In stark contrast to this, *fmo1 pen1 syp122* plants were unable to create an encasement around the developing IPS when penetrated by *Bgh*. To rule out the possibility that the lack of these defense structures is due to a host cell death response, dead cells stained by Trypan blue were excluded from our observations. Similar to our observations on papilla formation, we also assessed the ability of these plants to accumulate $H_2O_2$ in response to successful penetration by *Bgh* using DAB. We found that in *fmo1 pen1 syp122* plants successful penetration by *Bgh* induced a diffuse, superficial accumulation of $H_2O_2$, but we did not observe encasement-like structures (*Figure 4D and E*). This supported our finding that plants lacking PEN1 and SYP122 failed to form encasements at sites of attack. To verify the requirement for PEN1 and SYP122 in forming papillae and encasements, we included observations of *pad4 sid2 pen1 syp122* and *amsh3 pen1 syp122* mutants, where, similar to *fmo1*, the autoimmune phenotype of *pen1*

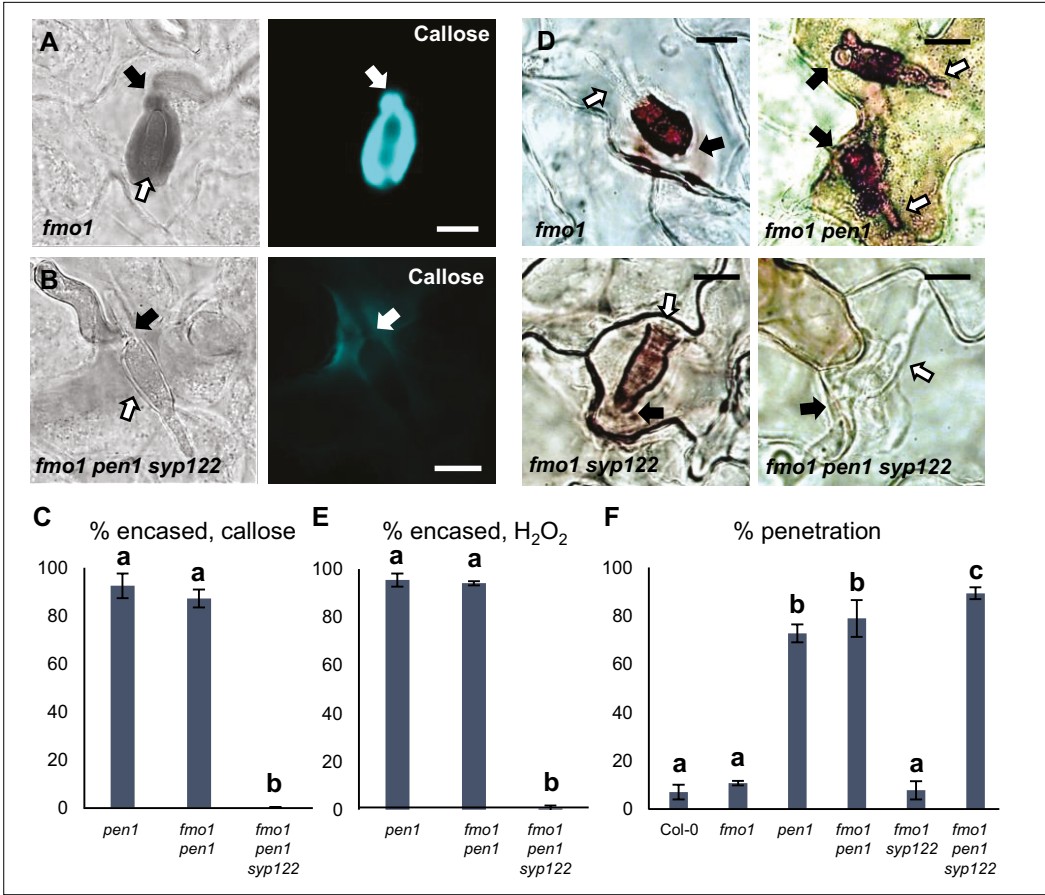

**Figure 4.** PEN1/SYP122 are required for encasement responses toward *Bgh*. (**A, B, D**) Accumulation of callose (**A, B**) or $H_2O_2$ (**D**) in response to *Bgh* attack (arrows) at penetrated attack sites. Open arrows point to the developing intracellular pathogenic structure (IPS). Bars = 10 μm. (**C, E**) Frequency of encasements in response to *Bgh* in penetrated cells as detected by staining for callose (**C**) or $H_2O_2$ (**E**). (**F**) Frequency of penetrations by *Bgh*. (**C, E, F**) All values are mean ± SD (n = 5 leaves per genotype). Different letters indicate significantly different values at p≤0.001 estimated using logistic regression.

The online version of this article includes the following video, source data, and figure supplement(s) for figure 4:

**Source data 1.** Source data for the graphs in *Figure 4C, E and F*.

**Figure supplement 1.** PEN1/SYP122 are required for encasement responses to *Bgh*.

**Figure supplement 2.** PEN1/SYP122 are required for papilla and encasement responses to *Bgh*.

**Figure 4—video 1.** Z-stack (callose and bright-field overlay) of successful penetration by *G. orontii* in *fmo1*.
https://elifesciences.org/articles/73487/figures#fig4video1

**Figure 4—video 2.** Z-stack (callose and bright-field overlay) of successful penetration by *G. orontii* in *fmo1 pen1 syp122*.
https://elifesciences.org/articles/73487/figures#fig4video2

**Figure 4—video 3.** Z-stack (callose and bright-field overlay) of successful penetration by *G. orontii* in *pad4 sid2 pen1 syp122*.
https://elifesciences.org/articles/73487/figures#fig4video3

**Figure 4—video 4.** Z-stack (callose and bright-field overlay) of successful penetration by *G. orontii* in *amsh3 pen1 syp122*.
https://elifesciences.org/articles/73487/figures#fig4video4

**Figure 4—video 5.** Z-stack (callose and bright-field overlay) of successful penetration by *Bgh* in *fmo1*.
https://elifesciences.org/articles/73487/figures#fig4video5

**Figure 4—video 6.** Z-stack (callose and bright-field overlay) of successful penetration by *Bgh* in *fmo1 pen1 syp122*.
https://elifesciences.org/articles/73487/figures#fig4video6

*Figure 4 continued on next page*

*Figure 4 continued*

https://elifesciences.org/articles/73487/figures#fig4video6

**Figure 4—video 7.** Z-stack (callose and bright-field overlay) of successful penetration by *Bgh* in *pad4 sid2 pen1 syp122.*

https://elifesciences.org/articles/73487/figures#fig4video7

**Figure 4—video 8.** Z-stack (callose and bright-field overlay) of successful penetration by *Bgh* in *amsh3 pen1 syp122.*

https://elifesciences.org/articles/73487/figures#fig4video8

*syp122* is attenuated (*Zhang et al., 2008*; *Schultz-Larsen et al., 2018*). As expected, fungal development and host cell responses in these lines were similar to that observed in *fmo1 pen1 syp122* plants (*Figure 4—figure supplement 2*, *Figure 4—videos 1–8*). As SYP122 appears to have a significant, albeit less important, role in papilla and encasement formation, we wondered if the complete absence of these defense structures would lead to an increase in penetration frequency as compared to the delayed papilla response described for the *pen1* mutant. We found a significant increase in fungal penetration frequency on *fmo1 pen1 syp122* plants when compared to *pen1* single and *fmo1 pen1* double mutant plants (*Figure 4F*). This supports our previous finding of a minor but significant PEN1-independent contribution to preinvasive immunity that relates to proper formation of the encasement (*Nielsen et al., 2017*). Taken together, these observations show that the *Bgh*-induced formation of papillae and encasements at attack sites is dependent on a functional overlap between the PEN1 and SYP122 syntaxins.

To further dissect the overlapping functions of PEN1 and SYP122 in immunity, we tested host cell responses to another, nonadapted fungal pathogen, namely, *C. destructivum* (an ascomycete fungal pathogen causing alfalfa anthracnose disease). The initial invasion strategy of *C. destructivum* is very similar to that of *Bgh* (*Latunde-Dada et al., 1997*). Nevertheless, while host cell responses include papilla formation, the highly effective preinvasive immunity of *Arabidopsis* toward nonadapted *Colletotrichum* species such as *C. destructivum* does not require PEN1 (*Shimada et al., 2006*; *Yang et al., 2014*). Moreover, attack by *C. destructivum* did not induce accumulation of GFP-PEN1, GFP-SYP122, or TET8-GFP at sites of attempted penetration (*Figure 5—figure supplement 1A–C*). Nonetheless, we found that plants lacking both PEN1 and SYP122 showed a dramatic decrease in the ability to form papillae at sites of unsuccessful penetration (*Figure 5A–C*). In contrast to *Bgh*, the host cell response to penetration by *C. destructivum* only rarely resulted in the formation of a detectable encasement. Nevertheless, penetrated host cells did respond by accumulating callose in and around the site of penetration. In *fmo1 pen1 syp122*, the responses varied from no detectable callose to a faint diffuse accumulation of callose at the penetration site (*Figure 5D and E*, *Figure 5—figure supplement 1D and E*). Moreover, the lack of host cell responses correlated with a remarkable increase in penetration frequency by *C. destructivum* on *fmo1 pen1 syp122* plants (*Figure 5F*, *Figure 5—figure supplement 1F and G*). Thus, preinvasive immunity toward *C. destructivum* requires a syntaxin function shared by PEN1 and SYP122, and this correlates with the lack of defense structure formation in response to attack.

Finding that PEN1 and SYP122 facilitate a preinvasive immunity that extends to a fungal pathogen other than powdery mildews, we speculated that this immunity could be a general defense mechanism effective against an even wider range of direct penetrating filamentous pathogens. One such pathogen is *P. infestans* (the potato blight pathogen), which, as an oomycete, is thought to have evolved completely independently of true fungi (*Lévesque, 2011*). Similar to the above observations for *C. destructivum*, preinvasive immunity toward *P. infestans* in *Arabidopsis* does not rely on a functional PEN1 (*Lipka et al., 2005*). Instead, *Rodriguez-Furlán et al., 2016* reported that loss of the root hair-specific SYP123 affects the penetration rates by *P. infestans*. However, SYP123 was suggested to regulate rhizobacterial priming of induced systemic resistance and is unlikely to be directly involved in the preinvasive response in leaves. We observed that attempted penetration by *P. infestans* did not lead to accumulation of either GFP-PEN1, GFP-SYP122, or TET8-GFP at the site of attack (*Figure 6—figure supplement 1A–C*). As expected, penetration attempts by *P. infestans* incited papilla formation in all genotypes, except for *fmo1 pen1 syp122* (*Figure 6A–C*). Successful penetration by *P. infestans* was often followed by host cell death. Nevertheless, we found that host cells expressing PEN1 and/or SYP122 could initiate a strong accumulation of callose and form an encasement around the IPS.

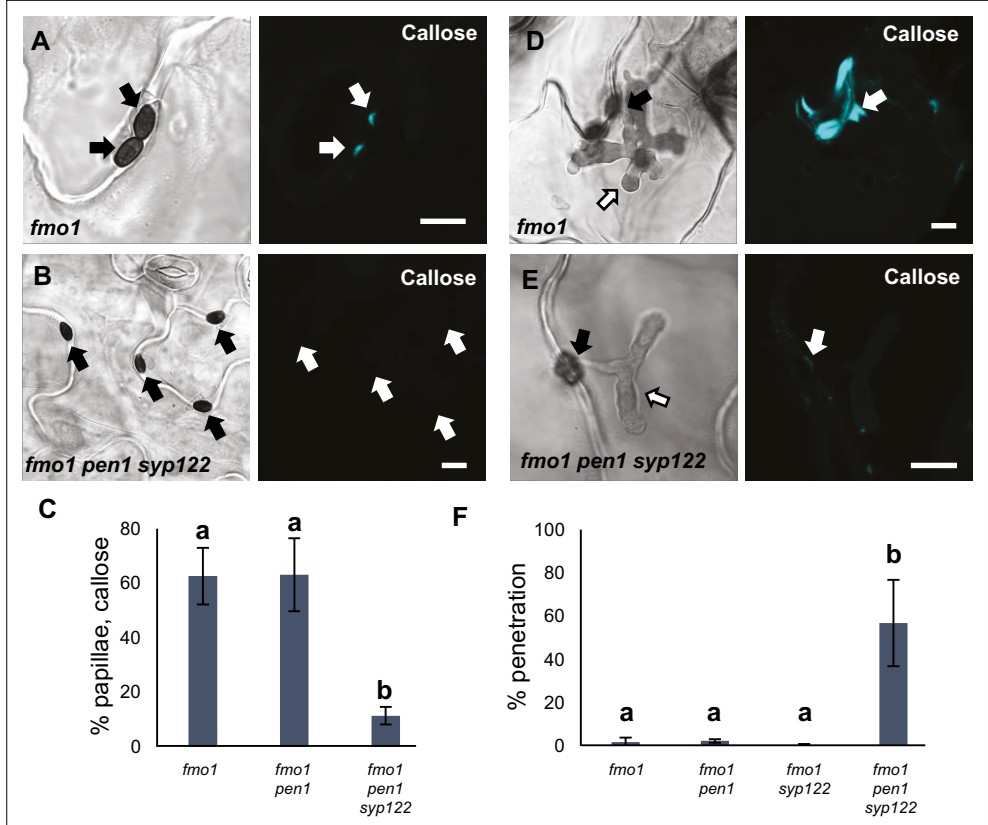

**Figure 5.** PEN1/SYP122 are required for preinvasive immunity toward *C. destructivum*. (**A, B, D, E**) Accumulation of callose in response to attack by melanized *C. destructivum* appressoria (arrows) at (**A, B**) nonpenetrated and (**D, E**) penetrated attack sites. Open arrows point to the developing intracellular pathogenic structure (IPS). Bars = 10 µm. (**C**) Frequency of papillae in response to *C. destructivum* in nonpenetrated cells. (**F**) Frequency of penetrations by *C. destructivum*. (**C, F**) All values are mean ± SD (n = 4 leaves per genotype). Different letters indicate significantly different values at p≤0.001 estimated using logistic regression.

The online version of this article includes the following source data and figure supplement(s) for figure 5:

**Source data 1.** Source data for the graphs in *Figure 5C and F*.

**Figure supplement 1.** PEN1/SYP122 are required for preinvasive immunity toward *C. destructivum*.

In the *fmo1 pen1 syp122* triple mutant, the penetrated cell, as well as neighboring cells, responded with a diffuse callose accumulation but encasement of the IPS was not seen (*Figure 6D and E*). Similar to our observations with *C. destructivum*, *fmo1 pen1 syp122* plants displayed elevated penetration frequencies when attacked by *P. infestans* (*Figure 6F*, *Figure 6—figure supplement 1D*). Surprisingly, we could not detect an increase in penetration events by *P. infestans* in the *syp123* mutant, even when in combination with *pen1*. Taken together, these findings show that PEN1 or SYP122 are required for papilla and encasement responses, which mediate immunity toward these phylogenetically distant, nonadapted filamentous pathogens. Moreover, it is striking that the functionality of PEN1 and SYP122 in immunity does not correlate with the ability of these syntaxins to visibly accumulate at attack sites.

PEN1 and SYP122 belong to the SYP12 clade of secretory syntaxins, which evolved during plant terrestrialization and is conserved in all land plants (*Sanderfoot, 2007*; *Slane et al., 2017*). To evaluate whether the immunity function of the SYP12 clade is also evolutionarily conserved among land plants, we investigated the response of the early diverging land plant *Marchantia* to attack by nonadapted filamentous pathogens. Consistent with previous findings, the germinating spores of *Bgh* were not able to properly differentiate a penetration hypha or appressorium on *Marchantia* thalli (*Figure 7—figure supplement 1A*; *Takikawa et al., 2014*). Similarly, most spores of *P. infestans* germinating on *Marchantia* seemed unable to identify a host cell and direct an attack. Nonetheless, occasional sites of attempted and successful penetrations were detected, to which the host cell responded by forming

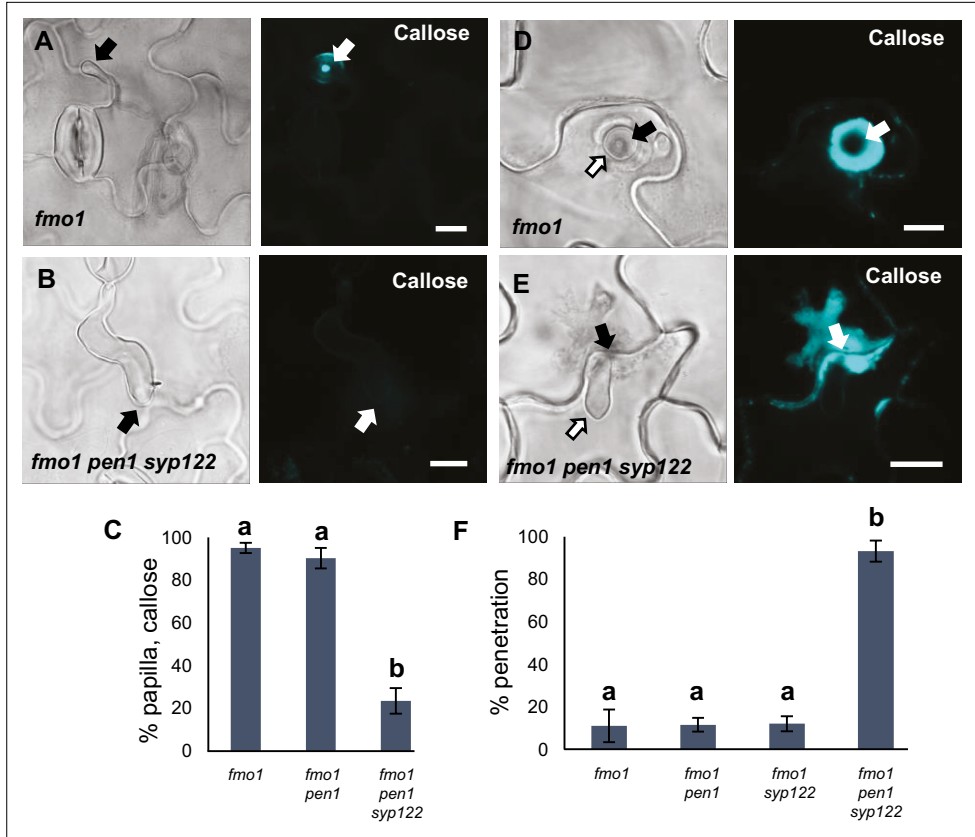

**Figure 6.** PEN1/SYP122 are required for preinvasive immunity toward *P. infestans*. (**A, B, D, E**) Accumulation of callose in response to *P. infestans* attack (arrows) at (**A, B**) nonpenetrated and (**D, E**) penetrated attack sites. Open arrows point to the developing intracellular pathogenic structure (IPS). Bars = 10 µm. (**C**) Frequency of papillae in response to *P. infestans* in nonpenetrated cells. (**F**) Frequency of penetrations by *P. infestans*. (**C, F**) All values are mean ± SD (n = 4 leaves per genotype). Different letters indicate significantly different values at p≤0.001 estimated using logistic regression.

The online version of this article includes the following source data and figure supplement(s) for figure 6:

**Source data 1.** Source data for the graphs in *Figure 6C and F*.

**Figure supplement 1.** PEN1/SYP122 are required for preinvasive immunity toward *P. infestans*.

papillae and encasement-like defense structures, respectively (*Figure 7A*, *Figure 7—figure supplement 1B*). Thus, neither *Bgh* nor *P. infestans* were suitable for studying nonhost interactions in *Marchantia*. In contrast, *C. destructivum* spores germinated and developed characteristic darkly melanized appressoria on the thallus surface. In response to attack, *Marchantia* epidermal cells formed callose-containing papillae similar to those found in *Arabidopsis* (*Figure 7B*). However, we never detected successful penetration of *Marchantia* cells by *C. destructivum*. A previous report that the *Marchantia* syntaxin MpSYP13B accumulates in response to colonization by *Phytophthora palmivora* prompted us to test whether candidate MpSYPs also accumulate in response to *C. destructivum* (*Carella et al., 2018*). However, despite inducing papillae, attack by *C. destructivum* did not induce focal accumulation of any of the four MpSYPs tested (*Figure 7C–F*; *Kanazawa et al., 2016*). Thus, the cellular responses of *Marchantia* to *C. destructivum* were similar to those that we observed in *Arabidopsis*.

In contrast to PEN1 and SYP122 in *Arabidopsis*, MpSYP12A seems to be essential in *Marchantia*, thereby precluding the possibility for studying preinvasive immunity in knockout lines (*Kanazawa et al., 2020*). Instead, to evaluate whether the immunity function of the SYP12 clade is evolutionarily conserved among land plants, we introduced GFP-fused versions of the two *Marchantia* SYP12s (i.e., MpSYP12A and MpSYP12B) into the *Arabidopsis pen1 syp122* mutant. GFP-MpSYP12A fully suppressed the necrotic and dwarfed phenotypes of *pen1 syp122* plants, while GFP-MpSYP12B only partially suppressed these phenotypes (*Figure 8—figure supplement 1A and B*). In contrast,

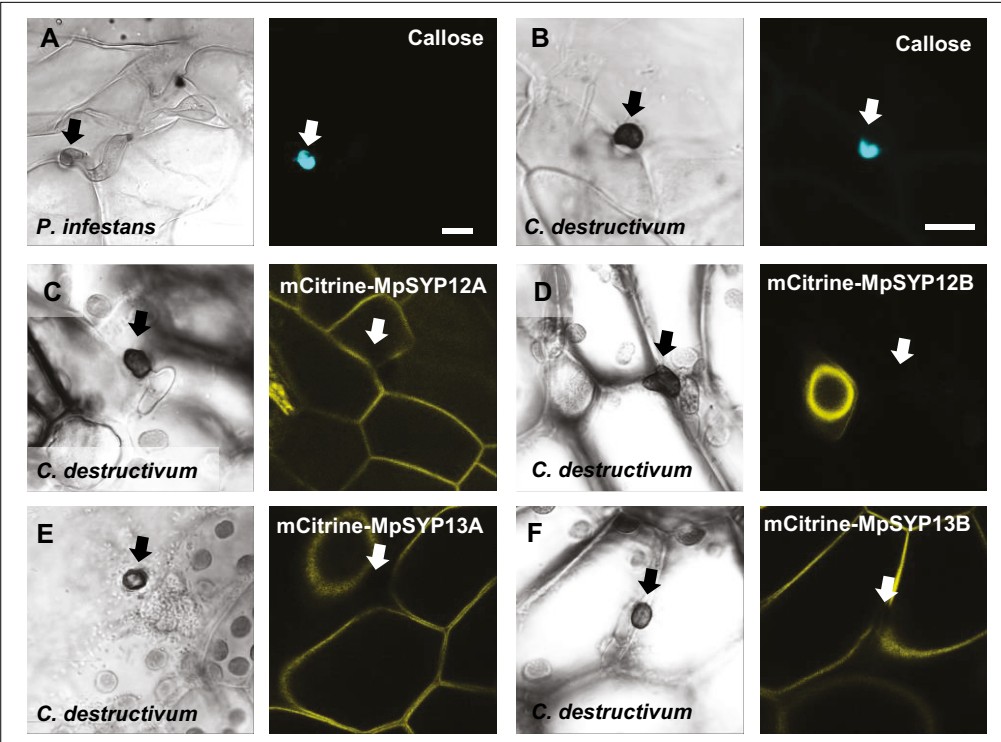

**Figure 7.** Responses in *Marchantia polymorpha* to filamentous pathogens. (**A, B**) Accumulation of callose in *Marchantia* in response to attack (arrows) by (**A**) *P. infestans* or (**B**) *C. destructivum* in nonpenetrated cells. Note that the frequency of *P. infestans* spores that attack a host cell is very low. (**C–F**) mCitrine signal of MpSYPs in response to attack (arrows) by *C. destructivum* in nonpenetrated cells. We did not observe any successful penetrations in *Marchantia* by *C. destructivum*. Bars = 10 µm.

The online version of this article includes the following figure supplement(s) for figure 7:

**Figure supplement 1.** Responses in *Marchantia polymorpha* to filamentous pathogens.

expression of either of the two SYP13 clade members, MpSYP13A and MpSYP13B, failed to rescue the *pen1 syp122* mutant. This suggests that MpSYP12A shares the PEN1/SYP122 overlapping function and that MpSYP12B does so incompletely. Because MpSYP12A complements the growth phenotype of *pen1 syp122* at the macroscopic level, we decided to test how these plants react to attack by *Bgh*. We found that MpSYP12A completely, and MpSYP12B partially, restored both papilla and encasement responses (*Figure 8A and B*, *Figure 8—figure supplement 2*, *Figure 8—figure supplement 3*). As explained above, the papillary accumulation of GFP-PEN1 in response to attack by *Bgh* is also seen for ROR2 in barley and highlights a likely conserved functionality in plant defense against powdery mildews (*Bhat et al., 2005*). In contrast, neither GFP-SYP122 nor the two GFP-MpSYP12s accumulated in papillae at sites of attempted penetration (*Figure 8C*, *Figure 8—figure supplement 4*). However, all the tested SYPs accumulated in the encasement matrix at sites of successful penetration (*Figure 8D*, *Figure 8—figure supplement 5*). Interestingly, preinvasive immunity toward *Bgh* was not restored by GFP-MpSYP12A. Instead, the penetration frequency of *Bgh* was similar to that of plants expressing GFP-SYP122 (*Figure 8E*). Thus, MpSYP12A resembles SYP122, and not PEN1, both in its localization and functionality. In support of this, GFP-MpSYP12A fully restored immunity toward both *P. infestans* and *C. destructivum* (*Figure 8F*, *Figure 8—figure supplement 6A and B*). Also, when tested with the adapted filamentous pathogens *G. orontii* and *Colletotrichum higginsianum*, penetration frequencies were similar to plants expressing PEN1 and SYP122 (*Figure 8—figure supplement 6C and D*). As described for PEN1 and SYP122, focal accumulation of GFP-MpSYP12A or GFP-MpSYP12B was not observed at unsuccessful attack sites by either *C. destructivum* or *P. infestans* (*Figure 8—figure supplement 7*). Taken together, these results reveal a conserved SYP12 clade functionality that has survived 470 My of independent evolution. Moreover, our findings suggest that

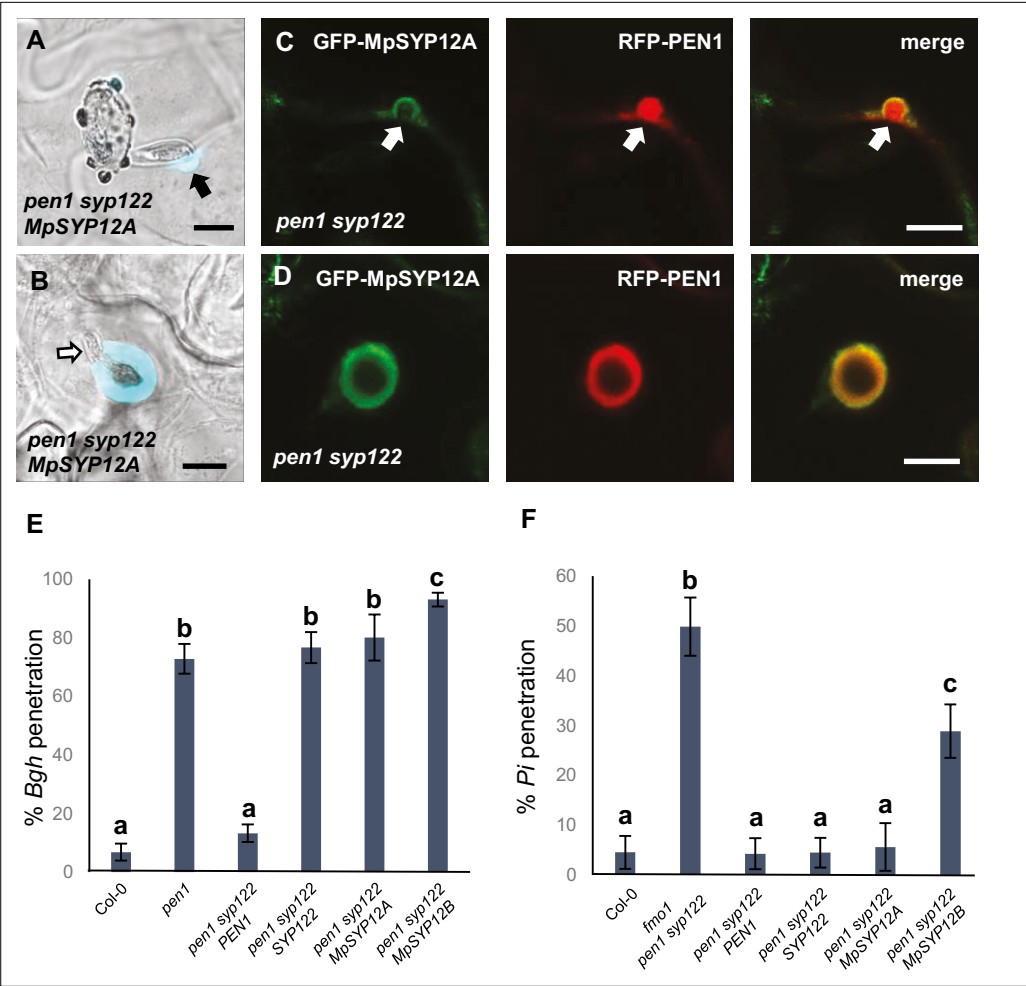

**Figure 8.** *Marchantia* syntaxins rescue papilla and encasement responses in *Arabidopsis*. (**A, B**) Accumulation of callose in response to attack (arrow) by *Bgh* in (**A**) nonpenetrated and (**B**) penetrated cells. Open arrow points to the developing intracellular pathogenic structure (IPS). (**C, D**) Localization of MpSYP12A and PEN1 in response to attack (arrows) by *Bgh* in (**C**) nonpenetrated and (**D**) penetrated cells. Bars = 10 µm. (**E, F**) Frequency of penetration by (**E**) *Bgh* and (**F**) *P. infestans*. All values are mean ± SD (n = 4 leaves per genotype). Different letters indicate significantly different values at p≤0.001 estimated using logistic regression.

The online version of this article includes the following source data and figure supplement(s) for figure 8:

**Source data 1.** Source data for the graphs in *Figure 8E and F*.

**Figure supplement 1.** *Marchantia* syntaxins are functional in *Arabidopsis*.

**Figure supplement 2.** *Marchantia* syntaxins rescue papilla responses.

**Figure supplement 3.** *Marchantia* syntaxins rescue encasement responses.

**Figure supplement 4.** Localization of SYP12 clade members at *Bgh* attack sites.

**Figure supplement 5.** Localization of SYP12 clade members at *Bgh* penetration sites.

**Figure supplement 6.** *Marchantia* SYP12s restore immunity in *Arabidopsis* toward *C. destructivum*.

**Figure supplement 7.** Localization of *Marchantia* SYP12 clade members at attack sites.

the PEN1-dependent immunity toward powdery mildew fungi evolved at a later stage of land plant evolution.

## Discussion

We have revealed a highly conserved role for the SYP12 clade of plant syntaxins in mediating the formation of pre- and postinvasive defense structures and provide immunity to highly diverged

filamentous pathogens. *Arabidopsis* plants lacking both PEN1 and SYP122 were incapable of forming papillae and encasements in response to attack by the nonadapted pathogens *Bgh*, *C. destructivum*, and *P. infestans*, as based on staining for callose and $H_2O_2$, resulting in highly elevated penetration frequencies. Instead, these plants responded to attack with diffuse and unfocused accumulations of callose and $H_2O_2$. It is noteworthy that although the *pen1 syp122* double mutant lacks these defense structures, epidermal cells are still able to respond to pathogen-derived signals in a polarized fashion, as seen in responses to the nonpenetrative primary germ tube of *Bgh*. This underlines that neither the general ability to secrete callose, generate $H_2O_2$, nor the underlying mechanism needed for polarized secretion are dependent on PEN1 and SYP122. Instead, we suggest that the PEN1/SYP122-dependent papillae and encasements are defense structures that enable a focused secretion, specifically aimed at blocking the ingress of filamentous pathogens. In support of this, knockdown of StSYR1 in potato, a homolog of PEN1, results in reduced papilla formation in response to *P. infestans* (*Eschen-Lippold et al., 2012*). A more indirect role of syntaxins in preinvasive immunity is the regulation of rhizobacterial priming of induced systemic resistance by the root hair-specific SYP123 (*Rodriguez-Furlán et al., 2016*). The induction of systemic resistance could affect the speed by which PEN1/SYP122-dependent papillae/encasements form in response to attack. However, as we failed to see an effect of SYP123, we could not pursue this question further. We are unsure of the reason for this discrepancy between our results and those reported by *Rodriguez-Furlán et al., 2016*, but we speculate that differences in the soil composition and microbiome could play a role in this. Instead, we have shown here that loss of papillae/encasements correlates with the loss of general preinvasive immunity. Yet, it is difficult to explain how these cell wall depositions would prevent invasion by pathogens secreting cell wall degrading enzymes. Also, we find it unlikely that accumulation of callose, $H_2O_2$, syntaxins, or even TET8 would have a direct effect on the pathogen. This is supported by our observations that neither PEN1, SYP122, nor TET8 focally accumulate in response to attack by *C. destructivum* or *P. infestans*, despite the formation of papillae. Instead, our findings reinforce the idea that papillae and encasements serve as defense structures that contain a complex mixture of exosomes, toxic metabolites, and/or antimicrobial enzymes and peptides that mediate immunity (*Kwon et al., 2008*). It is likely that

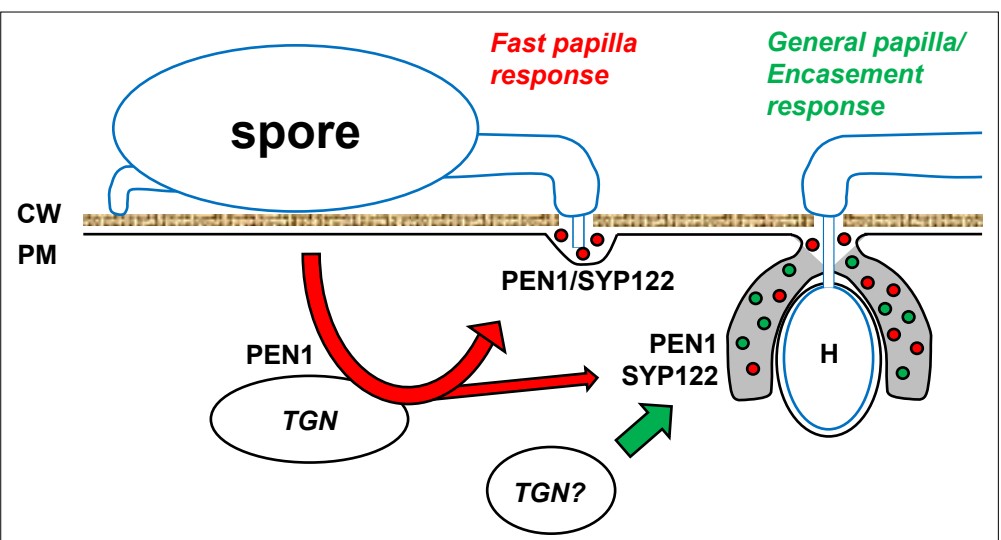

**Figure 9.** Schematic model of immunity against filamentous pathogens mediated by PEN1 and SYP122. In response to attack by filamentous pathogens, cargo for the papilla/encasement is received at the plasma membrane (PM) by PEN1 or SYP122 and secreted in between the plant cell wall (CW) and the PM . For pre-invasive immunity against powdery mildew fungi, PEN1 is required at the TGN to enable recycling of cargo that facilitates a fast papilla response (shown in red). The slower, general papilla/encasement response (shown in green) does not require rapid recycling at the TGN using PEN1. Instead, this response likely involves a specialization of the conventional secretory pathway or an alternative recycling pathway where either PEN1 or SYP122 are required at the PM to receive vesicular cargo for the papilla/encasement formation. Likely, the fast papilla response pathway also contributes to formation of the encasement resulting in a mixture of cargo (such as exosomes and anti microbial metabolites) transported by both pathways.

these cargo molecules are delivered in a PEN1- and SYP122-dependent manner to the site of attack to form papillae/encasements and thus prevent pathogen ingress.

Previous studies on the interaction between *Arabidopsis* and *Bgh* have revealed that preinvasive immunity correlates with the timely formation of papillae. This fast response likely requires the recycling of papilla material in a process involving PEN1 and GNOM (*Nielsen et al., 2012*). In contrast, although SYP122 also locates to the plasma membrane and interacts with the same SNARE partners as PEN1, *syp122* mutants show a wild-type response to *Bgh* (*Assaad et al., 2004*; *Pajonk et al., 2008*). Previous studies have shown that PEN1 and SYP122 mediate the secretion of distinct cargo subsets, which may explain their differing contributions to preinvasive immunity (*Waghmare et al., 2018*). Based on the work presented here, it is now evident that PEN1 and SYP122 functionally overlap in the formation of papillae and encasements. This could be explained by PEN1 serving two functions in preinvasive immunity. The first mediates a fast mobilization of papilla material, while the second function is shared with SYP122 and contributes to a more general papilla and encasement response to filamentous pathogens (*Figure 9*). An alternative explanation would be that, although redundant with SYP122, PEN1 somehow plays a more significant role in preinvasive immunity against *Bgh*. The fast mobilization of papilla material is likely related to the ability of PEN1 to shuttle between the plasma membrane and *trans*-Golgi network, which is less prominent for SYP122 (*Hansen and Nielsen, 2017*; *Nielsen and Thordal-Christensen, 2012*; *Nielsen and Thordal-Christensen, 2013*). Consequently, PEN1, and less so SYP122, could function at the *trans*-Golgi network, receiving vesicles from the plasma membrane that mediate the fast GNOM-dependent recycling of papilla material, warding off the powdery mildew fungus. We envisage that the redundant function of PEN1 and SYP122 is at the plasma membrane, where either of the two syntaxins receive the membrane material and vesicular cargo required for papilla/encasement formation. In contrast to the fast-acting, PEN1-dependent papilla response explained above, the slow-forming papilla uses either PEN1 or SYP122 already present at the site of attack in the plasma membrane and does not necessitate a fast recycling of the syntaxin itself. This could help to explain why PEN1 and SYP122 do not accumulate at sites of attack by *C. destructivum* and *P. infestans,* although both syntaxins clearly mediate preinvasive immunity. The pathway that delivers cargo for the slow-forming papilla is also likely to be involved in delivering cargo for the encasement because the formation of neither is affected by the lack of PEN1. Therefore, we speculate that the differences in papilla content based on observations of GFP-PEN1, GFP-SYP122, and TET8-GFP result from the involvement of distinct pathways for delivering cargo vesicles to the papilla. Alternatively, as the accumulation of some papilla cargos is triggered by only a subset of pathogens, the host cell might be able to differentiate the response according to the attacker. Nonetheless, these pathways share a common requirement for PEN1 or SYP122 to mediate vesicle fusion with the plasma membrane at pathogen attack sites. In addition, because some spores fail to penetrate, even in the complete absence of papillae and encasements, preinvasive immunity toward filamentous pathogens could also involve other molecular pathways that act independently of PEN1 and SYP122. In this regard, preinvasive immunity mediated by PEN2 and PEN3, as well as the resistance gained by loss of MLO, is of particular interest since both pathways seem to function independently of PEN1 (*Lipka et al., 2005*; *Stein et al., 2006*; *Kuhn et al., 2017*). It remains to be elucidated if these preinvasive immunity pathways also function independently of SYP122.

The SYP12 clade of plant secretory syntaxins is thought to have evolved during colonization of the land and is conserved in all land plants so far examined (*Slane et al., 2017*). Because terrestrialization presented enormous challenges to plants, the SYP12 clade was suggested to have acquired specialized roles in secretion needed to facilitate this transition (*Sanderfoot, 2007*). This is consistent with the previously identified functions of PEN1 (powdery mildew resistance), SYP123 (root hair tip growth), and SYP124 and SYP125 (pollen tube tip growth) all being associated with specialized forms of polarized secretion (*Collins et al., 2003*; *Slane et al., 2017*; *Ichikawa et al., 2014*). In addition, PEN1 promotes K$^+$ uptake by direct interaction with the RYxxWE motif found in a subset of Kv channels (*Karnik et al., 2017*). However, powdery mildews, root hairs, and pollen tubes did not exist until long after the onset of plant terrestrialization. Also, the interaction motifs needed for Kv channel interaction are restricted to vascular plants (*Karnik et al., 2017*). Therefore, this knowledge gives little insight into the original function of the SYP12 clade. Our discovery that SYP12A from *Marchantia* can complement SYP122, but not PEN1, in *Arabidopsis* reveals a conserved syntaxin functionality spanning 470 My of independent evolution. In accordance with this, MpSYP12A showed a localization behavior identical

to that of SYP122. Therefore, the functionality of the original SYP12 clade syntaxins is likely to have been similar to that of MpSYP12A and SYP122. At a later stage of plant evolution, PEN1 has acquired an additional specialization needed for powdery mildew resistance, which could explain its distinctive subcellular localization (*Nielsen and Thordal-Christensen, 2012*; *Nielsen and Thordal-Christensen, 2013*). Interestingly, in *Marchantia*, MpSYP12A locates to the plasma membrane and seems vital for cell plate formation, whereas MpSYP12B locates to a type of oil body unique to liverworts (*Kanazawa et al., 2016*; *Kanazawa et al., 2020*). This indicates yet another functional specialization of the SYP12 clade similar to that which occurred in *Arabidopsis*.

We suggest that MpSYP12A and SYP122 represent the original SYP12 clade functionality in land plants, which provides a highly conserved and effective broad-spectrum immunity toward filamentous pathogens. Before plant terrestrialization, the land was already colonized by filamentous fungi that were able to degrade cell walls and to utilize living or dead plant tissue as a nutrient source (*Chang et al., 2015*). Recent findings suggest that even before terrestrialization plants used NLRs to activate their defense responses (*Gao et al., 2018*). Activation of plant NLRs leads to a fast, programmed cell death response, which does not provide immunity against fungi growing on dead or dying plant tissue (*Wang et al., 2019a*; *Wang et al., 2019b*; *Horsefield et al., 2019*; *Wan et al., 2019*). Instead, the papilla/encasement defense structures would have conferred an effective defensive barrier against attack by the early saprophytic filamentous fungi already present on the land. Moreover, it is generally accepted that plant colonization of the land was aided by, or dependent on, mycorrhiza-like fungi (*Pirozynski and Malloch, 1975*; *Redecker et al., 2000*; *Heckman et al., 2001*). These mutualistic interactions are thought to have resulted from ancestral land plants being able to 'tame' early saprophytic filamentous fungi (*Chang et al., 2015*; *Berbee et al., 2017*). In this context, hyphal encasement may have provided a tool for plants to restrict and manipulate the intracellular growth of these fungi. We envisage that the SYP12 syntaxins enabled early land plants to regulate fungal entry and colonization, and that over time this leads to the mutually beneficial interactions that facilitated terrestrialization.

Adapted filamentous pathogens are able to overcome preinvasive immunity in their respective host plants, but not in other plant species. Our finding that plants could rely on a common membrane trafficking pathway to block pathogen entry should lead to new insights into how successful pathogens defeat this immunity by means of effector proteins (*Jones et al., 2016*). A future goal will be to identify the membrane trafficking targets of such effectors and to exchange those between hosts and nonhosts to generate a preinvasive immunity that pathogens cannot overcome.

## Materials and methods

**Key resources table**

| Reagent type (species) or resource | Designation | Source or reference | Identifiers | Additional information |
|---|---|---|---|---|
| Gene (*Arabidopsis thaliana*) | *pen1-1 syp122-1* | *Assaad et al., 2004* | | See Materials and methods |
| Gene (*A. thaliana*) | *fmo1-1 pen1-1 syp122-1* | *Zhang et al., 2008* | | See Materials and methods |
| Gene (*A. thaliana*) | *amsh3-4 pen1-1 syp122-1* | *Schultz-Larsen et al., 2018* | | See Materials and methods |
| Gene (*A. thaliana*) | *sid2 pad4 pen1 syp122* | *Zhang et al., 2008* | | See Materials and methods |
| Gene (*A. thaliana*) | TET8-GFP | *Boavida et al., 2013* | | See Materials and methods |

### Plant, fungal, and oomycete growth

Plants of *Arabidopsis thaliana* and *M. polymorpha* (Tak-1) were grown at 21°C, with 8 hr of light at 125 micro Einstein s$^{-1}$ m$^{-2}$ per day. All mutant plant lines were reported previously (*Collins et al., 2003*; *Assaad et al., 2004*; *Zhang et al., 2008*; *Rodriguez-Furlán et al., 2016*; *Schultz-Larsen et al., 2018*). The barley powdery mildew fungus, *Bgh* (isolate C15) and *Arabidopsis* powdery mildew, *G. orontii*, were propagated on barley and *Arabidopsis*, respectively. *C. destructivum* isolate LARS 709 and *C. higginsianum* isolate IMI 349063A were cultured on Mathur's medium containing glucose (2.8 g), $MgSO_4$-$7H_2O$ (1.22 g), $KH_2PO_4$ (2.72 g), Oxoid Mycological peptone (2.18 g), and agar (30 g) in 1 L of deionized water at 25°C in the dark. The *P. infestans* (field isolate 2017-DK-31-03 and SW1_A1) was cultured on rye sucrose (2%) agar supplemented with β-sitosterol (50 mg/L), pimaricin (10 mg/L), and ampicillin (100 mg/L) at 18°C in the dark (*Tzelepis et al., 2020*).

## Infection studies

Spores from *Bgh* and *G. orontii* were blown evenly onto the leaves of 4-week-old *Arabidopsis* plants and kept at 21°C. For *Colletotrichum*, spores were harvested in water from a 7–10-day-old culture. Spores were adjusted to a final concentration of $5 \times 10^5$ or $1 \times 10^6$ spores/mL and sprayed evenly onto the leaves of 5-week-old *Arabidopsis* plants. Following inoculation, the plants were sealed inside plastic bags and kept at 21°C. Zoospores of *P. infestans* were released from sporangia by addition of cold water and incubation at 4°C for 90 min. The zoospore suspension was diluted to $5 \times 10^5$ or $1 \times 10^6$ spores/mL and added to the abaxial side of attached leaves from 4-week-old *Arabidopsis* plants. The leaves were kept in a closed Petri dish on water-soaked filter paper at 21°C.

For scoring penetration success, leaf material was Trypan blue stained at 24 hr post inoculation for *Bgh* or *P. infestans*, 48 hr post inoculation for *G. orontii,* and 72 hr post inoculation for *C. destructivum* (*Keogh et al., 1980*). Penetration was determined by the presence of a developing IPS inside epidermal cells using light microscopy. To avoid the possibility that cells undergoing programmed cell death would be restricted in their encasement response to IPS, these cells were excluded from our observations based on Trypan blue staining. Once the penetration status for a particular event was determined, only then was the callose response investigated. For staining of live fungal structures, infected leaves were vacuum infiltrated with propidium iodide (10 µM) for 2 hr. Callose staining was performed using 0.01% Aniline blue in 1 M glycine (adjusted to pH 9.5 using NaOH) and visualized by UV epifluorescence. All experiments were repeated at least three times with similar results. Data from the individual studies described above represent discrete variables since it was recorded whether or not a certain event had taken place (e.g., whether or not a spore caused penetration). Consequently, these data were analyzed by logistic regression, assuming a binomial distribution (corrected for over-dispersion when present) (*Collett, 1991*). Hypotheses were rejected at $p < 0.001$. All data were analyzed by PC-SAS (release 9.4; SAS Institute, Cary, NC).

## RNA isolation and quantitative PCR

Total RNA was isolated from *A. thaliana* tissue using Monarch Total RNA Miniprep Kit (NEB). cDNA was synthesized using ProtoScript II First Strand cDNA Synthesis Kit (NEB) and an oligo(dT) primer. Transcript quantification was performed using the 5× HOT FIREPol EvaGreen qPCR Mix Plus (Solis BioDyne) on a Stratagene MX3000P real-time PCR detection system. Gene-specific primers are described below. We used the At4g26410 transcript as internal control.

    REF1_F2 TGAAGCTGCTGATTTGATGGA
    REF1_R2 CTCTAAGCTTGATAGCATCCCTTC
    PR1-F2 GAACACGTGCAATGGAGTTTG
    PR1-R2 CACTTTGGCACATCCGAGTCT

## Constructs

Open reading frames (ORFs) of PEN1 and SYP122 were amplified by PCR from cDNA, and the amplified products were subcloned into pDONR221 (Invitrogen). Along with pENTR clones containing MpSYP12A and MpSYP12B (*Kanazawa et al., 2016*), PEN1 and SYP122 were cloned into pUBIN-GFP, and PEN1 also in pB7WGR2 (*Karimi et al., 2002*; *Grefen et al., 2010*), destination vectors according to the manufacturer's instructions. For cloning, the following primers were used:

    attB1-for GGGGACAAGTTTGTACAAAAAAGCAGGCTAC
    attB2-rev GGGGACCACTTTGTACAAGAAAGCTGGGTC
    B1-PEN1-for AAAAAGCAGGCTACATGAACGATTTGTTTTCC
    B2-PEN1-rev AGAAAGCTGGGTCTCAACGCAATAGACGCCTTGC
    B1-SYP122-for AAAAAGCAGGCTACATGAACGATCTTCTCTCC
    B2-SYP122-rev AGAAAGCTGGGTCTTAGCGTAGTAGCCGCCG

## Fluorescent protein, propidium iodide, and callose detection using confocal microscopy

Samples were examined using a ×63 water immersion lens mounted on a Leica CLSM TCS SP5 confocal microscope. For detection and localization of the fluorophores and stains, GFP and propidium iodide were excited at 488 nm and detected between 500 and 520 nm, and 555 and 615 nm, respectively.

Aniline blue-stained callose was excited at 405 nm and detected between 435 and 485 nm. Imaging data were collected at the Center for Advanced Bioimaging (CAB) Denmark, University of Copenhagen. Projections of serial confocal sections, image overlays, and contrast enhancement were performed using image processing software (GIMP 2).

## Acknowledgements

We would like to thank Takashi Ueda for providing *Marchantia polymorpha* lines and cDNA clones of MpSYPs (*Kanazawa et al., 2016*). Also to Bent Jørgen Nielsen and Christina Dixelius for providing *Phytophthora infestans*. We are grateful to Christine Strullu-Derrien for fruitful discussions on the evolution of plant-fungus interactions and to Hans Thordal-Christensen, Stefan Wenkel, Sebastian Marquardt, Morten Petersen, and Pierre-Marc Delaux for useful comments on the manuscript. This project was supported by grants from the Villum Foundation (MEN, VKR023502), Independent Research Fund Denmark, Technical and Production Sciences (MEN, 6111-00524B), Novo Nordisk Foundation (MEN, NNF19OC0056457), Agence Nationale de la Recherche (RJO, ANR-17-CAPS-0004-01), and China Scholarship Council (ML, No. 201906300075).

## Additional information

### Funding

| Funder | Grant reference number | Author |
| --- | --- | --- |
| Villum Foundation | VKR023502 | Mads E Nielsen |
| Independent Research Fund Denmark | Technical and production series 6111-00524B | Mads E Nielsen |
| Novo Nordisk Fonden | NNF19OC0056457 | Mads E Nielsen |
| Agence Nationale de la Recherche | ANR-17-CAPS-0004-01 | Richard J O'Connell |
| China Scholarship Council | No. 201906300075 | Mengqi Liu |

The funders had no role in study design, data collection and interpretation, or the decision to submit the work for publication.

### Author contributions

Hector M Rubiato, Investigation, Methodology, Writing – review and editing; Mengqi Liu, Investigation, Writing – review and editing; Richard J O'Connell, Methodology, Writing – original draft, Writing – review and editing; Mads E Nielsen, Conceptualization, Funding acquisition, Investigation, Methodology, Supervision, Validation, Visualization, Writing – original draft, Writing – review and editing

### Author ORCIDs

Mads E Nielsen http://orcid.org/0000-0001-6170-8836

### Decision letter and Author response

Decision letter https://doi.org/10.7554/eLife.73487.sa1
Author response https://doi.org/10.7554/eLife.73487.sa2

## Additional files

### Supplementary files

• Transparent reporting form

### Data availability

All data generated or analysed during this study are included in the manuscript and supporting file; Source data files have been provided for Figures 3, 4, 5, 6 and 8.

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
