## [Editor Report]

The study provides evidence that PEN1 and SYP122 regulate structures important for defense against filamentous pathogen infection. The formation of papillae and encasement of haustoria both appear to be ancient defense mechanisms in land plants. The findings that PEN1 and its close homolog SYP122 play an overlapping role in pre- and post-invasive immunity against cell-wall penetrating filamentous pathogens advance our understanding of defense mechanisms against filamentous pathogens.

---

## [Decision Letter]

**Decision letter after peer review:**

[Editors’ note: the authors submitted for reconsideration following the decision after peer review. What follows is the editors’ comments after the first round of review.]

Thank you for submitting the paper "Plant SYP12 syntaxins mediate an evolutionarily conserved general immunity to filamentous pathogens" for consideration by *eLife*. Your article has been reviewed by 2 peer reviewers, including Jian-Min Zhou as the Reviewing Editor and Reviewer #1, and the evaluation has been overseen by a Senior Editor.

We are sorry to say that, after consultation with the reviewers, we have decided that this work will not be considered further for publication by *eLife*.

The reviewers judged that the experimental data are not sufficient to support the main conclusions. For instance, the claim that PEN1 has two separate functions with one specifically acting at the early stage of powdery mildew infection and one acts as a general mechanism against filamentous pathogens is not fully supported by the data. In addition, potential complications from autoimmunity are not ruled out, which hamper interpretation of the data.

[Editors’ note: the authors were encouraged for revision following their appeal, as described below.]

Thank you for submitting the rebuttal concerning our previous decision on your article "Plant SYP12 syntaxins mediate an evolutionarily conserved general immunity to filamentous pathogens". The prior decision was based on the reviewers' assessment that the claims are not fully supported by the data, and that the outcome of the experiments was anything but certain.

The reviewers and editors have since carefully read your explanation and planned revisions. I am happy to say that the reviewers are now in support of inviting revision. This evaluation has been overseen by a Reviewing Editor and Jürgen Kleine-Vehn as the Senior Editor.

Essential revisions:

1) A "specific" role of PEN1 in early Bgh response but not response to C. destructivum and P. infestans needs to be supported by careful study on the latter pathogens. If not, it is necessary to tone down the statement. Also, the statement that the role of PEN1 is executed by distinct trafficking mechanism is not convincing. A scenario in which PEN1 just plays the same but more significant role compared to SYP122 may be considered as an alternative to explain the genetic phenotypes observed. The drawing of the model proposed in Figure 9 is a bit confusing and it only depicts papilla but not the haustorial encasement.

2) The possibility that fmo1pen2syp122 has autoimmunity can hamper the conclusions drawn. It is necessary to show that autoimmunity is largely suppressed in these seedlings. For example, they do not constitutive express defense genes. Other mutants such as pen1syp122pad4sid2 (Zhang 2008) and pen1syp122amsh3 (Schultz-Larsen, 2018) should be used for the tests, as the autoimmunity phenotypes of these mutants seemed to be more significantly attenuated.

3) For Figure 1, because different fungal penetrations and differentiation of haustorium structures (including those in the same host cell) may occur at different levels, z-stack images showing no encasement despite differentiation of haustorium are needed. Similar questions also apply to other Figure 2 to Figure 4.

4) Since callosic papillae and haustorial encasements still form in Col-0 when infected by adapted powdery mildew such as G. orontii (Meyer, D., et al., Plant J, 2009, p986-999), the authors are advised to also use G. orontii or a similar isolate to challenge the

pen1syp122pad4sid2 and/or pen1syp122amsh3, or fmo1pen1syp122 to verify the results. Using such an adapted powdery mildew can ensure the formation of haustoria thus enabling reliable detection of both papillae and haustorial encasement.

5) In Figure 2, the IPS indicated by a dotted line is not clear. Again, it would be nice to have it stained by propidium iodide. Also, I wonder if the frequency of the focal accumulation of TET8-GFP around IPS in Col-0 is 100% and that in pen1syp122 is 0%. If it is not the case, some quantitative data to reflect the situation can help.

6) Figure 3, the quality of H2O2 staining images are poor.

7) It is not stated how successful penetration is scored.

*Reviewer #1:*

In this study the authors show that PEN1 and its close paralog SYP122 are required for pre-invasive (papillae formation) and invasive immunity (encasement of haustoria) against non-adapted powdery mildew fungal infection. Importantly, papillae formation on plants challenged with C. destructivum and P. infestans also required PEN1 and SYP122, suggesting that defense structures determined by the PEN1 and SYP122 operate against diverse filamentous pathogens. In addition, the authors provide strong evidence that the function specified by PEN1/SYP122 also exist in M. polymorpha, indicating an ancient evolution of the pre-invasive/invasive structures. Overall, the study makes a strong case that PEN1 and SYP122 play a crucial role in the biogenesis of ancient defense structures.

While much of the main conclusions is probably correct, there are certain statements not entirely supported by the data.

1. In several places the authors seem to claim that PEN1 and SYP122 are required for papillae formation and encasement of haustoria, but not callose deposition (Bgh infection). In other places, they show that pen1 syp122 fmo1 plants are defective in not only papillae but also callose deposition before penetration (C. destructivum and P. infestans infection). Can you really separate callose deposition and papillae formation?

2. The authors suggest that PEN1, but not SYP122 is required for pre-invasive immunity, whereas SYP122 and PEN2 are both required for invasive immunity (Figure 3). It seems more plausible that PEN1 and SYP122 are functionally redundant.

3. The authors suggest that auto-immunity in pen1 syp122 is likely caused by NLR activation (line 92), citing the requirement of SA and FMO1 for this autoimmunity. Have the authors tested requirement of EDS1, which would provide better support for the possibility. This is particularly important when the authors try to exclude interference from cell death (lines 184-186).

4. Figure 3E, "encasement of H2O2". It is difficult to comprehend how hydrogen peroxide is "encased". I think this suggest a defect in ROS burst. Together with the observed defect in callose deposition, the data seem to suggest a role of PEN1 and SYP122 beyond defense structures (papillae). I suggest the authors test PAMP-induced callose deposition and ROS burst in the fmo1 pen1 syp122 mutant.

5. Figure S3G, disease symptoms of C. destructivum-infected fmo1 pen1 syp122. Is this really disease symptoms or just increased cell death upon infection? Have the authors tested fungal biomass in the infected plants?

6. Lines 131-133, "the pen1 syp122…seems to be highly sensitive to signal released by Bgh". Have the authors tested sensitivity of the double mutant to PAMPs?

7. Lines 206-207 and 227-228. Bgh, but not C. destructivum or P. infestans, induced PEN1, SYP122, and TET8 accumulation at the site of infection. It is necessary to discuss what makes the difference.

8. Figures S6B and S7B lack the pen1 syp122 control.

*Reviewer #2:*

In this manuscript, the authors investigated the role of PEN1 and its close homolog SYP122 in pre- and post-invasive immunity reflected by the formation of papilla and haustorial encasement at the point of the attempted/successful penetration from non-adapted powdery and other filamentous pathogens. The authors collected genetic, cytological and cell biological data to show that these two syntaxins are required for the formation of the induced callosic defense structures. They further suggested that the two syntaxins function redundantly and differently. PEN1 may possess two different functions: the first is to mediate rapid delivery of papilla materials to powdery mildew penetration site, which is unique and not shared by SYP122, and the second is to work along with SYP122 to enable formation of a more general papilla/encasement in response to filamentous pathogens, and the latter is evolutionarily conserved as SYP12a from Marchantia can compensate the loss of SYP122.

My concerns/suggestion for improvement are as follows:

1) The delayed formation of papillae in pen1 plants in response to Bgh was observed and published before. Based on this, it is OK to propose that PEN1 has a distinctive role for early papilla response. But I am not sure if the authors have done similar time-course studies to see if the papilla formation in pen1 plants are also delayed in response to other filamentous pathogens. If not done yet, it is a bit early to infer that this PEN1-dependent rapid response is specific to powdery mildew.

2) Likewise, I am not completely convinced that PEN1 has two separable functions that may be executed via different trafficking mechanisms. A scenario in which PEN1 just plays the same but more significant role compared to SYP122 may be considered as an alternative to explain the genetic phenotypes observed. The drawing of the model proposed in Figure 9 is a bit confusing and it only depicts papilla but not the haustorial encasement.

3) For the data concerning the papillae/encasements in Figure 1 and many other figures, I would like to get some clarification from the authors.

a. I understand that the authors used small seedlings of pen1syp122 or fmo1pen2syp122 mutant plants that have not shown necrosis yet for the experiments. However, the timing of the activation of autoimmunity cannot be simply judged by the lack of necrotic lesions to the naked eye. In other words, the small seedlings of these genotypes may have strong constitutive defenses. This may raise a question as to whether Bgh spore germination and subsequent development is (partially) inhibited due to the defenses constitutively activated in the pen1syp122 double and even fmo1pen1syp122 triple (unless the authors had shown that the autoimmunity is largely suppressed in this mutant background).

b. Specifically, in Figure 1D, F and G, the authors observed large callosic deposits induced by the non-invasive primary germ tube, which were not observed in Col-0. This is almost counterintuitive. Any explanation? The sporeling shown in D,F seems smaller and/or irregularly shaped. I wonder if this is representative of all or majority sporelings found on leaves of pen1syp122. If so, the aberrant development could be indeed due to the autoimmunity activated in pen1syp122. The lack of callosic encasement may be due to no/rare formation of primitive haustoria (or IPS as called by the authors) from Bgh in leaf cells of pen1syp122. Staining the fungal structure using propidium iodide (plus vacuum) can help clarify. Also because different fungal penetrations and differentiation of haustorium structures (including those in the same host cell) may occur at different levels, hence z-stack images showing no encasement despite differentiation of haustorium can be more convincing. Similar questions may be asked about other figure (Figure 2 to Figure 4).

c. For this issue, I also wonder why the authors did not use pen1syp122pad4sid2 (Zhang 2008) and pen1syp122amsh3 (Schultz-Larsen, 2018) mutants for the tests, as the autoimmunity phenotypes of these mutants seemed to be more significantly attenuated.

d. Since callosic papillae and haustorial encasements still form in Col-0 when infected by adapted powdery mildew such as G. orontii (Meyer, D., et al., Plant J, 2009, p986-999), perhaps the authors may also use G. orontii or a similar isolate to challenge the pen1syp122pad4sid2 and/or pen1syp122amsh3, or fmo1pen1syp122 to verify the results. Using such an adapted powdery mildew can ensure the formation of haustoria thus enabling reliable detection of both papillae and haustorial encasement.

e. In Figure 2, the IPS indicated by a dotted line is not clear. Again, it would be nice to have it stained by propidium iodide. Also, I wonder if the frequency of the focal accumulation of TET8-GFP around IPS in Col-0 is 100% and that in pen1syp122 is 0%. If it is not the case, some quantitative data to reflect the situation can help.

f. Based on the images in Figure 3, H2O2 staining did not seem to be very effective, because the brownish staining was very weak even in fmo1 and fmo1pen1. So, the H2O2 data in Figure 3 do not seem to be super convincing to me. Also, I wonder how the author assessed successful penetration. I guess successful penetration in fmo1 or fmo1pen1 and fmo1syp122 can be indicated by the formation of callosic encasement, but it would be very difficult to do so in fmo1pen1syp122, as there was little/no callosic encasement as claimed by the authors. I wonder how the authors can reliably assess it. The same question can be asked for % penetration data in other figures.

---

## [Author Response]

[Editors’ note: The authors appealed the original decision. What follows is the authors’ response when encouraged for revision.]

Essential revisions:1) A "specific" role of PEN1 in early Bgh response but not response to C. destructivum and P. infestans needs to be supported by careful study on the latter pathogens. If not, it is necessary to tone down the statement. Also, the statement that the role of PEN1 is executed by distinct trafficking mechanism is not convincing. A scenario in which PEN1 just plays the same but more significant role compared to SYP122 may be considered as an alternative to explain the genetic phenotypes observed. The drawing of the model proposed in Figure 9 is a bit confusing and it only depicts papilla but not the haustorial encasement.

We have toned down the claim that PEN1 has a specific (second) function in response to Bgh in contrast to SYP122 (see below for further elaboration on this point). We have included a revised version of the model that includes the formation of the encasement (Figure 9).

2) The possibility that fmo1pen2syp122 has autoimmunity can hamper the conclusions drawn. It is necessary to show that autoimmunity is largely suppressed in these seedlings. For example, they do not constitutive express defense genes. Other mutants such as pen1syp122pad4sid2 (Zhang 2008) and pen1syp122amsh3 (Schultz-Larsen, 2018) should be used for the tests, as the autoimmunity phenotypes of these mutants seemed to be more significantly attenuated.

We have performed qRT-PCR showing that loss of FMO1 markedly reduces the expression of PR1 (Figure S2B). We have also included observations on pen1 syp122 pad4 sid2 and pen1 syp122 amsh3 as suggested, which were entirely consistent with those obtained on fmo1 pen2 syp122 plants (Figure S4).

3) For Figure 1, because different fungal penetrations and differentiation of haustorium structures (including those in the same host cell) may occur at different levels, z-stack images showing no encasement despite differentiation of haustorium are needed. Similar questions also apply to other Figure 2 to Figure 4.

As suggested, we have included videos of z-stacks to make this point clearer.

4) Since callosic papillae and haustorial encasements still form in Col-0 when infected by adapted powdery mildew such as G. orontii (Meyer, D., et al., Plant J, 2009, p986-999), the authors are advised to also use G. orontii or a similar isolate to challenge the pen1syp122pad4sid2 and/or pen1syp122amsh3, or fmo1pen1syp122 to verify the results. Using such an adapted powdery mildew can ensure the formation of haustoria thus enabling reliable detection of both papillae and haustorial encasement.

The manuscript now includes observations of G. orontii on pen1 syp122 and rescued lines (i.e. fmo1, pad4 sid2 and amsh3) (Figure S1, 2 and 4).

5) In Figure 2, the IPS indicated by a dotted line is not clear. Again, it would be nice to have it stained by propidium iodide. Also, I wonder if the frequency of the focal accumulation of TET8-GFP around IPS in Col-0 is 100% and that in pen1syp122 is 0%. If it is not the case, some quantitative data to reflect the situation can help.

The images in Figure 2 has been replaced and now shows the IPS stained with propidium iodide as requested.

6) Figure 3, the quality of H2O2 staining images are poor.

Images showing H2O2 staining have been replaced (Figure 3 and 4).

7) It is not stated how successful penetration is scored.

The description for evaluating successful penetration has been clarified (Lines 662-668).

Reviewer #1:In this study the authors show that PEN1 and its close paralog SYP122 are required for pre-invasive (papillae formation) and invasive immunity (encasement of haustoria) against non-adapted powdery mildew fungal infection. Importantly, papillae formation on plants challenged with C. destructivum and P. infestans also required PEN1 and SYP122, suggesting that defense structures determined by the PEN1 and SYP122 operate against diverse filamentous pathogens. In addition, the authors provide strong evidence that the function specified by PEN1/SYP122 also exist in M. polymorpha, indicating an ancient evolution of the pre-invasive/invasive structures. Overall, the study makes a strong case that PEN1 and SYP122 play a crucial role in the biogenesis of ancient defense structures.While much of the main conclusions is probably correct, there are certain statements not entirely supported by the data.1. In several places the authors seem to claim that PEN1 and SYP122 are required for papillae formation and encasement of haustoria, but not callose deposition (Bgh infection). In other places, they show that pen1 syp122 fmo1 plants are defective in not only papillae but also callose deposition before penetration (C. destructivum and P. infestans infection). Can you really separate callose deposition and papillae formation?

We use callose as a marker for highlighting papillae and encasements, and there is a tight correlation between them that requires PEN1 or SYP122. Nevertheless, it is clear that plants lacking PEN1 and SYP122 are still capable of secreting callose. We think there is a difference between PEN1/SYP122-dependent callose secretion seen for papillae and encasements, and callose secretion seen in other responses. We have tried to make this distinction much clearer in the revised manuscript.

2. The authors suggest that PEN1, but not SYP122 is required for pre-invasive immunity, whereas SYP122 and PEN2 are both required for invasive immunity (Figure 3). It seems more plausible that PEN1 and SYP122 are functionally redundant.

Indeed, we suggest that PEN1 and SYP122 are functionally redundant in the broad immunity that impedes attack by C. destructivum and P. infestans. On top of this, PEN1 has an additional role in mediating immunity towards the powdery mildew. We have tried to better differentiate their roles in responses to mildew and non-mildew pathogens in the revised manuscript.

3. The authors suggest that auto-immunity in pen1 syp122 is likely caused by NLR activation (line 92), citing the requirement of SA and FMO1 for this autoimmunity. Have the authors tested requirement of EDS1, which would provide better support for the possibility. This is particularly important when the authors try to exclude interference from cell death (lines 184-186).

Previously a number of knock-outs (including EDS1) have been tested for their ability to rescue the pen1 syp122 autoimmunity phenotype. Yet, in our hands the fmo1 mutation rescues better than the others. As requested, we have included observations on papilla and encasement formation in other rescuing lines (see below) to provide further support for our claims.

4. Figure 3E, "encasement of H2O2". It is difficult to comprehend how hydrogen peroxide is "encased". I think this suggest a defect in ROS burst. Together with the observed defect in callose deposition, the data seem to suggest a role of PEN1 and SYP122 beyond defense structures (papillae). I suggest the authors test PAMP-induced callose deposition and ROS burst in the fmo1 pen1 syp122 mutant.

Similar to callose, we envisage that localized H2O2 accumulation in papillae is a consequence of papilla/encasement formation and not the other way around. Please note that in pen1 syp122, despite the lack of papillae/encasements there is still production of callose and H2O2 in response to successful penetrations (and sometimes even at unsuccessful attack sites). However, in the absence of a proper defense structure, i.e. papilla/encasement, this results in an unfocused accumulation of callose and H2O2. Thus, plants lacking PEN1 and SYP122 still produce callose and H2O2 (indicating a normal response) but fail to direct it to the site of attack.

5. Figure S3G, disease symptoms of C. destructivum-infected fmo1 pen1 syp122. Is this really disease symptoms or just increased cell death upon infection? Have the authors tested fungal biomass in the infected plants?

We think that the reviewer is correct in the interpretation that the macroscopic phenotype could include increased cell death upon infection and not only necrotic disease symptoms as such. This has been corrected in the revised manuscript (Figure text Figure S5 and S12).

6. Lines 131-133, "the pen1 syp122…seems to be highly sensitive to signal released by Bgh". Have the authors tested sensitivity of the double mutant to PAMPs?

We feel that the hypothesis that pen1 syp122 could be more sensitive to PAMPs, while very interesting, lies beyond the scope of this manuscript.

7. Lines 206-207 and 227-228. Bgh, but not C. destructivum or P. infestans, induced PEN1, SYP122, and TET8 accumulation at the site of infection. It is necessary to discuss what makes the difference.

In line 352 of the original manuscript we did discuss why the different pathogens lead to different responses in papilla content. However, we have now elaborated further on this in the revised manuscript (Lines 370-374).

8. Figures S6B and S7B lack the pen1 syp122 control.

The pen1 syp122 mutant was omitted from these experiments due to the severe phenotype that makes it very difficult to obtain data comparable to the rescued lines. Since MpSYP12B gives an intermediate phenotype in regards to phenotype and papilla/encasement formation, we find that this line serves as a better control.

Reviewer #2:In this manuscript, the authors investigated the role of PEN1 and its close homolog SYP122 in pre- and post-invasive immunity reflected by the formation of papilla and haustorial encasement at the point of the attempted/successful penetration from non-adapted powdery and other filamentous pathogens. The authors collected genetic, cytological and cell biological data to show that these two syntaxins are required for the formation of the induced callosic defense structures. They further suggested that the two syntaxins function redundantly and differently. PEN1 may possess two different functions: the first is to mediate rapid delivery of papilla materials to powdery mildew penetration site, which is unique and not shared by SYP122, and the second is to work along with SYP122 to enable formation of a more general papilla/encasement in response to filamentous pathogens, and the latter is evolutionarily conserved as SYP12a from Marchantia can compensate the loss of SYP122.My concerns/suggestion for improvement are as follows:1) The delayed formation of papillae in pen1 plants in response to Bgh was observed and published before. Based on this, it is OK to propose that PEN1 has a distinctive role for early papilla response. But I am not sure if the authors have done similar time-course studies to see if the papilla formation in pen1 plants are also delayed in response to other filamentous pathogens. If not done yet, it is a bit early to infer that this PEN1-dependent rapid response is specific to powdery mildew.

We agree that we cannot, at this stage, state that the PEN1 dependent rapid response is specific to powdery mildews. We have modified this accordingly in the revised manuscript.

2) Likewise, I am not completely convinced that PEN1 has two separable functions that may be executed via different trafficking mechanisms. A scenario in which PEN1 just plays the same but more significant role compared to SYP122 may be considered as an alternative to explain the genetic phenotypes observed. The drawing of the model proposed in Figure 9 is a bit confusing and it only depicts papilla but not the haustorial encasement.

Evidence for two separable functions of PEN1 comes from previous work showing PEN1’s specific role in powdery mildew resistance, papilla accumulation, localization in response to BFA, interaction with Ion-channels and specificity in secretory cargo, and is as such not a new idea. However, we think that the reviewer is correct that at least the distinct role in response to Bgh could be explained by PEN1 being a more interactive syntaxin in comparison to SYP122 (i.e. doing the same function, but more efficiently and/or faster). We have incorporated this line of thought into the discussion (Lines 354-361), and made the model easier to comprehend (Figure 9).

3) For the data concerning the papillae/encasements in Figure 1 and many other figures, I would like to get some clarification from the authors.a. I understand that the authors used small seedlings of pen1syp122 or fmo1pen2syp122 mutant plants that have not shown necrosis yet for the experiments. However, the timing of the activation of autoimmunity cannot be simply judged by the lack of necrotic lesions to the naked eye. In other words, the small seedlings of these genotypes may have strong constitutive defenses. This may raise a question as to whether Bgh spore germination and subsequent development is (partially) inhibited due to the defenses constitutively activated in the pen1syp122 double and even fmo1pen1syp122 triple (unless the authors had shown that the autoimmunity is largely suppressed in this mutant background).

It is correct that the autoimmune reactions caused by the loss of PEN1 and SYP122 could be envisaged to hamper fungal development. However, from our experience this does not seem to be the case. Instead, penetration frequencies are even higher than the respective controls and in the fmo1 pen1 syp122 triple mutant (in which the autoimmunity is indeed largely suppressed, see Figure S1C-D) the haustoria even develop finger-like projections (similar to their normal development in the natural host, barley).

b. Specifically, in Figure 1D, F and G, the authors observed large callosic deposits induced by the non-invasive primary germ tube, which were not observed in Col-0. This is almost counterintuitive. Any explanation?

We are indeed puzzled by this observation, although it does show that the cell is able to deposit callose in the absence of PEN1 and SYP122. As we only observe these deposits in the non-rescued pen1 syp122 mutant, we think that this response could be related to the unsuppressed autoimmunity that somehow leads to a strong response to the primary germtube.

The sporeling shown in D,F seems smaller and/or irregularly shaped. I wonder if this is representative of all or majority sporelings found on leaves of pen1syp122. If so, the aberrant development could be indeed due to the autoimmunity activated in pen1syp122.

We have not observed any effect on spore development on leaves of pen1 syp122. Also, the penetration rates are always very high, which would indicate normal development of the spores.

The lack of callosic encasement may be due to no/rare formation of primitive haustoria (or IPS as called by the authors) from Bgh in leaf cells of pen1syp122. Staining the fungal structure using propidium iodide (plus vacuum) can help clarify. Also because different fungal penetrations and differentiation of haustorium structures (including those in the same host cell) may occur at different levels, hence z-stack images showing no encasement despite differentiation of haustorium can be more convincing. Similar questions may be asked about other figure (Figure 2 to Figure 4).

Following a successful penetration, the host cell in pen1 syp122 mutant plants most often initiates programmed cell death. We did worry that the lack of encasements in pen1 syp122 could be due to improper haustoria development or initiation of cell death (or even a combination of the two). Therefore, we made use of the fmo1 pen1 syp122 triple mutant (in which the autoimmunity is largely suppressed, see Figure S1C-D). We did succeed in producing images with a haustorium stained by propidium iodide which have replaced the old images in Figure 2. In addition, we have included videos of Z-stack projections, showing the fungal structure inside the host cell of pen1 syp122 and all rescuing lines (i.e. fmo1, pad4 sid2 and amsh3).

c. For this issue, I also wonder why the authors did not use pen1syp122pad4sid2 (Zhang 2008) and pen1syp122amsh3 (Schultz-Larsen, 2018) mutants for the tests, as the autoimmunity phenotypes of these mutants seemed to be more significantly attenuated.

As mentioned above, we have included pen1 syp122 pad4 sid2 and pen1 syp122 amsh3 rescuing lines to confirm the observations found in pen1 syp122 fmo1.

d. Since callosic papillae and haustorial encasements still form in Col-0 when infected by adapted powdery mildew such as G. orontii (Meyer, D., et al., Plant J, 2009, p986-999), perhaps the authors may also use G. orontii or a similar isolate to challenge the pen1syp122pad4sid2 and/or pen1syp122amsh3, or fmo1pen1syp122 to verify the results. Using such an adapted powdery mildew can ensure the formation of haustoria thus enabling reliable detection of both papillae and haustorial encasement.

Indeed G. orontii also induces encasement formation, although at a later stage and less frequently. We have included observations of successful penetrations by G. orontii as suggested by the reviewer.

e. In Figure 2, the IPS indicated by a dotted line is not clear. Again, it would be nice to have it stained by propidium iodide. Also, I wonder if the frequency of the focal accumulation of TET8-GFP around IPS in Col-0 is 100% and that in pen1syp122 is 0%. If it is not the case, some quantitative data to reflect the situation can help.

As described above, we have replaced the images in Figure 2. We have also included in the text that TET8-GFP seems to be a reliable marker for the encasement in Col-0 and we also indicate the frequency of extracellular accumulation TET8-GFP in encasements (Lines 159163).

f. Based on the images in Figure 3, H2O2 staining did not seem to be very effective, because the brownish staining was very weak even in fmo1 and fmo1pen1. So, the H2O2 data in Figure 3 do not seem to be super convincing to me.

We agree that in the images the staining for H2O2 does seem weak. Yet, in our hands the H2O2 staining gives reliable results. We have replaced with new images in which the H2O2 staining is more visible (Figure 3 and 4).

Also, I wonder how the author assessed successful penetration. I guess successful penetration in fmo1 or fmo1pen1 and fmo1syp122 can be indicated by the formation of callosic encasement, but it would be very difficult to do so in fmo1pen1syp122, as there was little/no callosic encasement as claimed by the authors. I wonder how the authors can reliably assess it. The same question can be asked for % penetration data in other figures.

Successful penetration is always scored on the presence of a haustorium (or IPS), which can be determined using bright-field or DIC microscopy without difficulty. Once the penetration status for a particular event has been determined, only then do we investigate the callose response. We have made this point more clearly in the revised manuscript (Lines 662-668).